# Learning Dynamic Graph Representation of Brain Connectome with Spatio-Temporal Attention

**Byung-Hoon Kim** [*]
Department of Psychiatry
Institute of Behavioral Sciences in Medicine
College of Medicine, Yonsei University
egyptdj@yonsei.ac.kr

**Jong Chul Ye**
Department of Bio/Brain Engineering
Kim Jaechul Graduate School of AI
KAIST
jong.ye@kaist.ac.kr

**Jae-Jin Kim**
Department of Psychiatry
Institute of Behavioral Sciences in Medicine
College of Medicine, Yonsei University
jaejkim@yonsei.ac.kr

## Abstract

Functional connectivity (FC) between regions of the brain can be assessed by the degree of temporal correlation measured with functional neuroimaging modalities. Based on the fact that these connectivities build a network, graph-based approaches for analyzing the brain connectome have provided insights into the functions of the human brain. The development of graph neural networks (GNNs) capable of learning representation from graph structured data has led to increased interest in learning the graph representation of the brain connectome. Although recent attempts to apply GNN to the FC network have shown promising results, there is still a common limitation that they usually do not incorporate the dynamic characteristics of the FC network which fluctuates over time. In addition, a few studies that have attempted to use dynamic FC as an input for the GNN reported a reduction in performance compared to static FC methods, and did not provide temporal explainability. Here, we propose STAGIN, a method for learning *dynamic* graph representation of the brain connectome with spatio-temporal attention. Specifically, a temporal sequence of brain graphs is input to the STAGIN to obtain the dynamic graph representation, while novel READOUT functions and the Transformer encoder provide spatial and temporal explainability with attention, respectively. Experiments on the HCP-Rest and the HCP-Task datasets demonstrate exceptional performance of our proposed method. Analysis of the spatio-temporal attention also provide concurrent interpretation with the neuro-scientific knowledge, which further validates our method. Code is available at https://github.com/egyptdj/stagin

## 1 Introduction

Neuroimaging modalities provide measurements of brain activity by capturing the signals of neural activity. Functional magnetic resonance imaging (fMRI) is a non-invasive imaging method that measures the blood-oxygen level dependence (BOLD) in order to estimate the neural activity of the whole brain over time [20]. Functional connectivity (FC) is defined as the degree of temporal

---

[*]This work was conducted while the first author was at KAIST

35th Conference on Neural Information Processing Systems (NeurIPS 2021).

correlation between regions of the brain. Based on the fact that these connectivities form networks that change over time, graph-based network analysis of brain connectome has been one of the key approaches to understanding how the brain works [7, 4, 43].

Graph neural networks (GNNs) are a type of deep neural networks that have recently been successful in learning the representation of graph-structured data [50]. The graph-structured nature of the brain has led to an increased interest in learning the reperesentation of the brain FC network with the GNNs. Learning the representation of the brain connectome can be linked to decoding trait or state from human brain signal measurements. Accordingly, the current trend in studies attempting to apply GNN to the brain connectome is to input the FC graph from either resting-state [25, 26, 2, 33, 23, 48, 49] or task fMRI data [29, 30, 32] and predict a particular phenotype of the subjects, such as gender [26, 2, 23, 22] or presence of a specific disease [26, 33, 29, 30, 32, 22]. While these studies have shown potential strengths and opportunities for learning the network representation of the brain, they also suggest limitations of current GNN-based methods.

One of the most common limitations with previous GNN-based FC network analysis methods is that most of them fail to take advantage of the dynamic properties of the FC network, which fluctuates over time. Incorporating the dynamic features of the FC network into the neuroimaging analysis has been an important direction in the field of functional neuroimaging [21, 37]. A work by [15] tried to address this issue by using the Spatial Temporal Graph Convolutional Network (ST-GCN) [54] model to incorporate dynamic features of the FC network. However, [15] reported lower accuracy than other non-dynamic GNN-based FC methods [23, 2] in the gender classification experiment, leaving a question about the effectiveness of the dynamic FC method. In addition, another limitation of the method is that no temporal explainability is provided from the model. This is a major drawback considering that the goal of applying GNNs to functional neuroimaging methods is not only to achieve high classification accuracy, but also to uncover the functional basis of the brain [23, 32]. Another recent work by [3], using GraphNets [5] and DiffPool [58] for the dynamic FC analysis, also suffers from the same limitations in terms of poor classification accuracy and lack of temporal explainability.

Here, we propose Spatio-Temporal Attention Graph Isomorphism Network (STAGIN) for learning the dynamic graph representation of the brain connectome with spatio-temporal attention. The proposed method exploits the temporal features of the dynamic FC network graphs to improve the classification accuracy of the model. In particular, we address the issue that the node features of the input dynamic graph should contain temporal information and concatenate encoded timestamp with the node features (Section 4.1). In addition, the proposed method includes novel attention-based READOUT modules (Section 4.2) and the Transformer encoder [46] (Section 4.3) in order to further improve the classification performance and provide spatial-temporal explainability at the same time. STAGIN achieves state-of-the-art performance with the Human Connectome Project (HCP) dataset [45] in gender classification for resting-state fMRI and task decoding for task fMRI. We inherit k-means clustering analysis of the resting-state dynamic FC [1] and general linear model (GLM) statistical mapping of task fMRI [14] for interpreting the spatio-temporal attention learned from STAGIN, which are widely accepted analysis methods for the fMRI data. The interpretation of the learned spatio-temporal attention replicates neuroscientific findings from previous large-scale fMRI studies in both resting-state and task fMRI, which further validates our proposed method.

Our work holds potential societal impact in that brain decoding methods can be linked to finding neural biomarkers of important phenotypes or diseases. However, potential negative impact related to privacy concerns that arise from abuse or misuse of accurate decoding methods should also be noted. Although our method is yet behind the decoding capability that can be abused or misused, our research cannot still be free from these ethical considerations.

## 2    Related works

### 2.1    Graph Neural Network on Dynamic Graphs

Many networks that arise around us are inherently dynamic, with changes in the existence of nodes and edges over time. Learning the representation of dynamic graphs has piqued the interest of researchers and has led to development of methods that can embed dynamic graphs using their time information [35]. Methods that incorporate attention for learning the representation of dynamic graphs have also been proposed [51, 40]. However, it is not easy to apply these techniques directly to the dynamic brain graphs because of the different inherent properties of the dynamic brain graphs that

do not include any addition or deletion of nodes and are sampled uniformly over time. Nonetheless, our work is inspired by these earlier studies, particularly for the encoding of temporal information and their concatenation to the node features, as proposed in Section 4.1 [51, 40].

## 2.2 Attention in Graph Neural Networks

Bringing attention to the GNNs is a topic that is being actively studied in the field of geometric deep learning [27]. One of the most successful uses of attention is to compute the attention at edges of the graph and scale the importance of the links when the features of the neighborhood node are aggregated [47, 6], often providing performance gain in learning the representation of input graphs. Another stream of applying attention to the GNNs comes with the motivation to define a pooling function on the graph domain. Since it is not straightforward to decide on what basis the coarsening should be carried out for graph structured data, works such as [16, 28, 38] have addressed this problem by selecting the nodes with top scores computed from projecting the node feature vectors into a learnable parameter vector, or from a GNN layer aggregated local graph features. Although the motivation may have been different, these graph pooling methods are closely related to the spatial attention modules that we propose in Section 4.2 in that they exploit learned relative scores across the vertices of the graph. While some works have already been aware that the appropriate use of node-wise attention can improve performance of downstream tasks [55, 11], we note that previous methods tend to score attention based on randomly initialized parameters or local graph structures which may be suboptimal for graph classification tasks that require taking the whole graph feature into account.

# 3 Theory

## 3.1 Problem definition

The goal of our study is to train a neural network

$$f : G_{\text{dyn}} \rightarrow \boldsymbol{h}_{G_{\text{dyn}}},$$

where $G_{\text{dyn}} = (G(1), ..., G(T))$ is the sequence of brain graphs with $T$ timepoints and $\boldsymbol{h}_{G_{\text{dyn}}} \in \mathbb{R}^D$ is the vector representation of the dynamic graph $G(t)$ with length $D$. The graph $G(t) = (V(t), E(t))$ at time $t$ is a pair of vertex set $V(t) = \{\boldsymbol{x}_1(t), ..., \boldsymbol{x}_N(t)\}$ of $N$ nodes and edge set $E(t) = \left\{ \{\boldsymbol{x}_i(t), \boldsymbol{x}_j(t)\} \mid j \in \mathcal{N}(i), i \in \{1, ...N\} \right\}$ where $\mathcal{N}(i)$ denotes the neighborhood of the vertex $i$. If $f$ learns to extract a disentangled representation of the dynamic brain graph $G_{\text{dyn}}$, then the classification of a certain phenotypic characteristic (e.g. gender) from $\boldsymbol{h}_{G_{\text{dyn}}}$ can be performed with a linear mapping as a downstream task. Another important consideration in this work is to ensure the explainability of the model $f$, being able to inform us which part of the brain at which timepoint was considered important when extracting the meaningful representation $\boldsymbol{h}_{G_{\text{dyn}}}$. Specifically, we formulate $f = q \circ g$ as a composition of the GNN $g$ and the Transformer encoder $q$, where $g$ outputs the set of graph representations $\boldsymbol{h}_{G(t)}$ from each timepoint and $q$ exploits self-attention to integrate $\boldsymbol{h}_{G(t)}$ into the final representation $\boldsymbol{h}_{G_{\text{dyn}}}$:

$$g : G_{\text{dyn}} \rightarrow (\boldsymbol{h}_{G(1)}, ..., \boldsymbol{h}_{G(T)}), \tag{1}$$

$$q : (\boldsymbol{h}_{G(1)}, ..., \boldsymbol{h}_{G(T)}) \rightarrow \boldsymbol{h}_{G_{\text{dyn}}}. \tag{2}$$

We will omit timepoint notation $(t)$ for brevity, whenever it is not of contextual importance.

## 3.2 Graph Isomorphism Network

The GNNs are generally composed of functions that (i) integrate the node features from its neighbors, and (ii) embed the integrated information with a nonlinear transformation to obtain the next layer node features. These functions are called AGGREGATE, and COMBINE functions, respectively, and the choice of these functions define many variants of the GNN,

$$\boldsymbol{a}_v^{(k)} = \texttt{AGGREGATE}^{(k)} \left( \left\{ \boldsymbol{h}_u^{(k-1)} : u \in \mathcal{N}(v) \right\} \right), \tag{3}$$

$$\boldsymbol{h}_v^{(k)} = \texttt{COMBINE}^{(k)} \left( \boldsymbol{h}_v^{(k-1)}, \boldsymbol{a}_v^{(k)} \right), \tag{4}$$

where $\boldsymbol{h}_v^{(k)}$ denotes the feature vector of node $v$ at layer $k$ and $\boldsymbol{h}_v^{(0)} := \boldsymbol{x}_v$.

The Graph Isomorphism Network (GIN) is a variant of the GNN suitable for graph classification tasks, which is known to be as powerful as the WL-test under certain assumptions of injectivity [52]. The GIN typically defines sum as the AGGREGATE and a multi-layer perceptron (MLP) with two layers as the COMBINE updating the node representation $\boldsymbol{h}_v^{(k)}$ at layer $k$ [52] by :

$$\boldsymbol{h}_v^{(k)} = \texttt{MLP}^{(k)}\Big( (1 + \epsilon^{(k)}) \cdot \boldsymbol{h}_v^{(k-1)} + \sum_{u \in \mathcal{N}(v)} \boldsymbol{h}_u^{(k-1)} \Big), \tag{5}$$

where $\epsilon$ is a learnable parameter initialized with zero. Equation (5) can be easily reformulated into the matrix form [23] by:

$$\boldsymbol{H}^{(k)} = \sigma\left( (\epsilon^{(k)} \cdot \boldsymbol{I} + \boldsymbol{A}) \boldsymbol{H}^{(k-1)} \boldsymbol{W}^{(k)} \right), \tag{6}$$

where

$$\boldsymbol{H}^{(k)} = \left[ \boldsymbol{h}_1^{(k)}, \cdots, \boldsymbol{h}_N^{(k)} \right] \in \mathbb{R}^{D \times N}$$

is the stack of node feature vectors, $\boldsymbol{I}$ is the identity matrix, $\boldsymbol{A}$ is the adjacency matrix between the node features, $\boldsymbol{W}$ is the network weights of the MLP, and $\sigma$ is the nonlinearity function.

The READOUT function takes the updated node features $\boldsymbol{h}_v^{(k)}$ to compute the representation of the whole graph:

$$\boldsymbol{h}_G^{(k)} = \texttt{READOUT}\Big( \{ \boldsymbol{h}_v^{(k)} \mid v \in G \} \Big). \tag{7}$$

In general, the READOUT function is defined simply as computing the sum or average of the input node features. This is equivalent to multiplication with the length $N$ pooling vectors $\boldsymbol{\phi}_{\text{sum}}^\top = [1, ..., 1]$ or $\boldsymbol{\phi}_{\text{mean}}^\top = [1/N, ..., 1/N]$ for the matrix form:

$$\boldsymbol{h}_G^{(k)} = \boldsymbol{H}^{(k)} \boldsymbol{\phi}_{\text{mean}}. \tag{8}$$

### 3.3  Encoder-decoder understanding of GNNs

Although formulating the GIN (5) as a combination of AGGREGATE and COMBINE function might not suggest its close relationship with convolutional neural networks (CNNs) at first glance, previous works by [23, 8] show that the matrix formulation of the GIN operation (6) can be thought of a CNN layer with shift operation of the convolution as the adjacency matrix $\boldsymbol{A}$. We extend the understanding of encoder-decoder CNN as a framelet expansion [57, 56] to the GIN to formulate node feature vectors $\boldsymbol{H}^{(k)}$ at layer $k$ with respect to the input node feature $\boldsymbol{x}_i$ as:

$$\text{Vec}\left( \boldsymbol{H}^{(k)} \right) = \boldsymbol{\Sigma}^{(k)} \boldsymbol{E}^{(k)\top} \cdots \boldsymbol{\Sigma}^{(1)} \boldsymbol{E}^{(1)\top} \boldsymbol{x}, \quad \text{where} \quad \boldsymbol{x} := \text{Vec}\left( [\boldsymbol{x}_1, \cdots, \boldsymbol{x}_N] \right) \tag{9}$$

where $\text{Vec}(\cdot)$ denotes the vectorization operation, and the $k$-th layer encoder matrix $\boldsymbol{E}^{(k)}$ is defined as

$$\boldsymbol{E}^{(k)} = \boldsymbol{W}^{(k)} \otimes (\epsilon^{(k)} \cdot \boldsymbol{I} + \boldsymbol{A}^T)$$

where $\otimes$ refers to the Kronecker product, and $\boldsymbol{\Sigma}^{(k)}$ is the diagonal matrix with values 1 or 0 depending on the activation pattern of the nonlinearity. Now, $\boldsymbol{\phi}_{\text{mean}}$ of equation (8) can be thought as the decoder at the $k$-th layer which yields the whole graph feature vector from the encoded node feature vectors.

**Proposition 1.** *The READOUT function $\boldsymbol{\phi}_{\text{mean}}$ in (8) generates a decoder with fixed constant bases.*

*Proof.* From the READOUT function (8), we have

$$\boldsymbol{h}_G^{(k)} = \text{Vec}\left( \boldsymbol{h}_G^{(k)} \right) = \text{Vec}\left( \boldsymbol{H}^{(k)} \boldsymbol{\phi}_{\text{mean}} \right) = \left( \boldsymbol{\phi}_{\text{mean}}^T \otimes \boldsymbol{I} \right) \text{Vec}\left( \boldsymbol{H}^{(k)} \right)$$

$$= \left( \boldsymbol{\phi}_{\text{mean}}^T \otimes \boldsymbol{I} \right) \boldsymbol{\Sigma}^{(k)} \boldsymbol{E}^{(k)\top} \cdots \boldsymbol{\Sigma}^{(1)} \boldsymbol{E}^{(1)\top} \boldsymbol{x}$$

Now let $\boldsymbol{b}_i$ and $\tilde{\boldsymbol{b}}_i$ denote the $i$-th column of the encoder matrix $\boldsymbol{E}^{(1)} \boldsymbol{\Sigma}^{(1)} \cdots \boldsymbol{E}^{(k)} \boldsymbol{\Sigma}^{(k)}$ and the decoder matrix $\left( \boldsymbol{\phi}_{\text{mean}}^T \otimes \boldsymbol{I} \right)$, respectively. Then, it is straight to obtain the following representation:

$$\boldsymbol{h}_G^{(k)} = \sum_i \langle \boldsymbol{b}_i, \boldsymbol{x} \rangle \tilde{\boldsymbol{b}}_i$$

Therefore, we can see that although the encoder basis $\boldsymbol{b}_i$ is a function of $\boldsymbol{x}$, the decoder basis $\tilde{\boldsymbol{b}}_i$ is a constant. $\qquad \square$

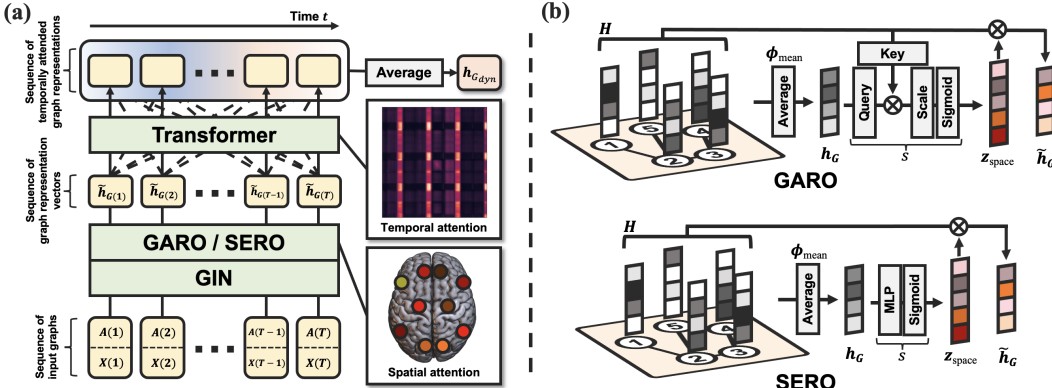

Figure 1: Schematic illustration of the proposed method. (a) Overall framework of the STAGIN. A sequence of dynamic graph is first input to the GIN followed by GARO or SERO which produces a sequence of spatially attended graph representation vectors $\tilde{\boldsymbol{h}}_{G(t)}$. Temporal attention is computed over $\tilde{\boldsymbol{h}}_{G(t)}$ and the temporally attended graph representations are averaged to generate the final representation $\tilde{\boldsymbol{h}}_{G_{\mathrm{dyn}}}$. (b) Attention-based READOUT modules. Both GARO and SERO compute spatial attention $\boldsymbol{z}_{\mathrm{space}}$ with global average-pooled graph feature $\boldsymbol{h}_G$ as prior.

We address the issue that the decoder being a constant function can restrict the expressivity of the neural network, and explore adaptive READOUT functions with attention in Section 4.2.

# 4 STAGIN: Spatio-Temporal Attention Graph Isomorphism Network

In this section, we discuss the details of our main contribution. Specifically, we propose STAGIN with two novel attention-based READOUT modules for learning the dynamic graph representation of the brain connectome (Figure 1).

## 4.1 Dynamic graph definition

The sequence of input dynamic FC graphs is constructed from 4D fMRI data with 3D voxels across time. The ROI-timeseries matrix $\boldsymbol{P} \in \mathbb{R}^{N \times T_{\max}}$ is extracted by taking the mean values within a pre-defined 3D atlas which consists of $N$ ROIs at each timepoint. Values of each ROI are standardized across time. Constructing dynamic FC matrix follows the sliding-window approach, where the temporal window of length $\Gamma$ is shifted across time with stride $S$ to generate $T = \lfloor T_{\max} - \Gamma/S \rfloor$ windowed matrices $\bar{\boldsymbol{P}}(t) \in \mathbb{R}^{N \times \Gamma}$ (Figure 2 (a)). The FC at time $t$ is defined as the correlation coefficient matrix $\boldsymbol{R}(t)$ of the windowed timeseries between $\bar{\boldsymbol{p}}_i(t)$ and $\bar{\boldsymbol{p}}_j(t)$:

$$R_{ij}(t) = \frac{\mathrm{Cov}(\bar{\boldsymbol{p}}_i(t), \bar{\boldsymbol{p}}_j(t))}{\sigma_{\bar{\boldsymbol{p}}_i}(t)\sigma_{\bar{\boldsymbol{p}}_j}(t)} \in \mathbb{R}^{N \times N},$$

where the subscript $i$ and $j$ are the row and column indices of $\bar{\boldsymbol{P}}(t)$, Cov denotes the cross covariance, and $\sigma_{\boldsymbol{p}}$ denotes the standard deviation of $\boldsymbol{p}$. The final binary adjacency matrix $\boldsymbol{A}(t) \in \{0,1\}^{N \times N}$ is obtained from the FC matrix $\boldsymbol{R}(t)$ by thresholding the top 30-percentile values of the correlation matrix as connected, and otherwise unconnected following [23]. Other thresholds for binarizing the correlation matrix are also experimented and the results are provided in the Appendix Section C.2.

Unlike the adjacency matrix $\boldsymbol{A}(t)$, conventional definition of node feature vectors $\boldsymbol{x}_v(t)$ at node index $v$ as coordinates [29], mean-activation [29, 15], or one-hot encoding [23], do not change over $t$, disregarding any temporal variation. To address this issue, we concatenate encoded timestamp $\eta(t) \in \mathbb{R}^D$ to the spatial one-hot encoding $\boldsymbol{e}_v$, followed by linear mapping with a learnable parameter matrix $\boldsymbol{W} \in \mathbb{R}^{D \times (N+D)}$ to define the input node feature,

$$\boldsymbol{x}_v(t) = \boldsymbol{W}[\boldsymbol{e}_v || \eta(t)]. \tag{10}$$

Here, the learnable timestamp encoder $\eta$ is a Gated Recurrent Unit (GRU) [9] which takes ROI-timeseries upto the endpoint of the sliding-window as the input. Both the vertex set $V(t)$ and the

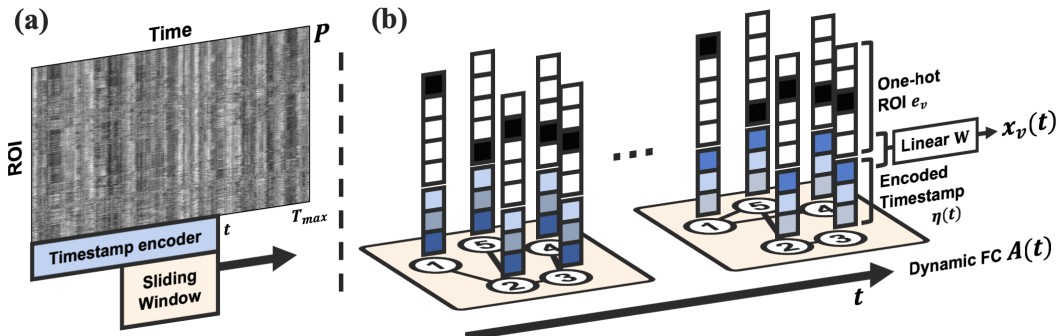

Figure 2: Defining the dynamic graph. (a) Scheme of extracting dynamic graph from ROI-timeseries matrix $\boldsymbol{P}$. (b) Example of a constructed dynamic graph.

edge set $E(t)$ of graph $G(t)$ now incorporates temporal information at time $t$. See Figure 2 for an illustration of the dynamic graph definition.

## 4.2 Spatial attention with attention-based READOUT

As suggested from Proposition 1, conventional READOUT function of GNN can be thought of as a fixed decoder that decodes whole-graph feature from the node features with no learnable parameters. We address this issue by incorporating attention to the READOUT function, which the attention here refers to the scaling coefficient across the nodes learned by the model. Specifically, the spatial attention vector $\boldsymbol{z}_{\text{space}}(t) \in [0,1]^N$ is computed by taking the $\boldsymbol{H}$ as a prior:

$$\boldsymbol{z}_{\text{space}} = s(\boldsymbol{H}), \tag{11}$$

$$\tilde{\boldsymbol{h}}_G = \boldsymbol{H}\boldsymbol{z}_{\text{space}}, \tag{12}$$

where $s : \mathbb{R}^{D \times N} \to [0,1]^N$ is the attention function and $\tilde{\boldsymbol{h}}_G$ denotes spatially attended graph representation $\boldsymbol{h}_G$. We propose two types of attention function $s(\cdot)$ for the attention-based READOUT, named Graph-Attention READOUT (GARO) and Squeeze-Excitation READOUT (SERO) inspired by the attention mechanisms of [46] and [19], respectively.

### 4.2.1 GARO: Graph-Attention READOUT

The GARO follows key-query embedding based attention of the Transformer [46]. However, the key embedding is computed from the matrix of node features $\boldsymbol{H}$, while the query embedding is computed from the vector of unattended graph representation $\boldsymbol{H}\boldsymbol{\phi}_{\text{mean}}$:

$$\boldsymbol{K} = \boldsymbol{W}_{\text{key}}\boldsymbol{H},$$
$$\boldsymbol{q} = \boldsymbol{W}_{\text{query}}\boldsymbol{H}\boldsymbol{\phi}_{\text{mean}},$$
$$\boldsymbol{z}_{\text{space}} = \text{sigmoid}\Big(\frac{\boldsymbol{q}^\top \boldsymbol{K}}{\sqrt{D}}\Big), \tag{13}$$

where $\boldsymbol{W}_{\text{key}} \in \mathbb{R}^{D \times D}$, $\boldsymbol{W}_{\text{query}} \in \mathbb{R}^{D \times D}$ are learnable key-query parameter matrices, $\boldsymbol{K} \in \mathbb{R}^{D \times N}$ is the embedded key matrix, and $\boldsymbol{q} \in \mathbb{R}^D$ is the embedded query vector.

### 4.2.2 SERO: Squeeze-Excitation READOUT

The SERO follows MLP based attention of the Squeeze-and-Excitation Networks [19]. However, attention from the *squeezed* graph representation does not scale the channel dimension, but the node dimension in SERO:

$$\boldsymbol{z}_{\text{space}} = \text{sigmoid}\Big(\boldsymbol{W}_2\sigma(\boldsymbol{W}_1\boldsymbol{H}\boldsymbol{\phi}_{\text{mean}})\Big), \tag{14}$$

where $\sigma$ is the nonlinearity function and $\boldsymbol{W}_1 \in \mathbb{R}^{D \times D}$, $\boldsymbol{W}_2 \in \mathbb{R}^{N \times D}$ are learnable parameter matrices. This type of spatial dimension squeeze-excitation module has been shown to improve

performance of the CNN models [41], but was not easily applicable to general graphs which may vary in number of nodes for each graph. We exploit the fact that the brain graphs have fixed number of nodes $N$ across participants based on the chosen atlas.

### 4.2.3 Orthogonal regularization

If we take a closer look at (8) and (12), computation of graph feature vector $\boldsymbol{h}_G$ from the node feature matrix $\boldsymbol{H}$ can also be viewed as reconstructing signal $\boldsymbol{h}_G$ from the basis frames $\boldsymbol{H}$ with vectors $\phi_{\text{mean}}$ and $\boldsymbol{z}_{\text{space}}$, respectively. While $\boldsymbol{z}_{\text{space}}$ provides further expressivity of the model with adaptive coefficients when compared to $\phi_{\text{mean}}$, we find it desirable to encourage the orthogonality of $\boldsymbol{H}$ as elaborated in the Appendix Section A. The orthogonal regularization $\mathcal{L}_{\text{ortho}}$ is defined as:

$$\mathcal{L}_{\text{ortho}} = \left\| 1/m \cdot \boldsymbol{H}^\top \boldsymbol{H} - \boldsymbol{I} \right\|_2, \tag{15}$$

where $m = \max(\boldsymbol{H}^\top \boldsymbol{H})$. The scaling term $1/m$ ensures the columns of the matrix $\boldsymbol{H}$ become orthogonal to each other with the same length, while not restricting the specific length that the column vectors should follow.

### 4.3 Temporal attention with Transformer encoder

For attention across time, we employ a single-headed Transformer encoder [46] upon the sequence of graph features $(\tilde{\boldsymbol{h}}_{G(1)}, ..., \tilde{\boldsymbol{h}}_{G(T)})$. The temporal attention can be measured by the self-attention weights $\boldsymbol{Z}_{\text{time}} \in [0, 1]^{T \times T}$ after the softmax function of the Transformer encoder. Per-layer dynamic graph representation $\boldsymbol{h}_{G_{\text{dyn}}}^{(k)}$ is computed by summing the temporally attended feature output from the Transformer encoder across time at each layers, where the final representation:

$$\boldsymbol{h}_{G_{\text{dyn}}} = \text{concatenate}(\{\boldsymbol{h}_{G_{\text{dyn}}}^{(k)} \mid k \in \{1, ..., K\}\}) \tag{16}$$

is the concatenation of dynamic graph representation of all $K$ layers following [53].

## 5 Experiment

### 5.1 Dataset

Publicly available[2] fMRI data from the HCP S1200 release [45] was used for our experiments. The data was collected from voluntary participants with informed consent and was fully anonymized. We constructed two datasets, the HCP-Rest and the HCP-Task, depending on whether the subject was resting or performing specific tasks during the acquisition of the image. The HCP-Rest dataset consisted of pre-processed and ICA denoised resting-state fMRI data [17], which the subjects were instructed to rest for 15 minutes during the data acquisition. We used first run data of the four sessions, and excluded data with short acquisition time with $T_{\text{max}} < 1200$. There were 1093 images finally included in the dataset, which consisted of 594 female and 499 male subjects. The gender of each subject served as the labels of the HCP-Rest dataset letting the number of classes $C = 2$. The HCP-Task consisted of pre-processed task fMRI data [17], which the subjects were instructed to perform specific tasks during data acquisition. For example in the "Motor" task fMRI, participants were told to perform one of the subtasks during the acquisition to make motor movements on one's left hand, left foot, right hand, right foot, or tongue. There were seven types of tasks including working memory, social, relational, motor, language, gambling, and emotion. After excluding the fMRI data with short acquisition time, there were 7450 images included in the dataset. The task type during the data acquisition served as the labels of the HCP-Task dataset, letting $C = 7$. A more detailed description of the experiment datasets with a note on the twin subjects of HCP can be found in the Appendix Section B.

### 5.2 Experimental settings

Experiments were performed on a workstation with two NVIDIA GeForce GTX 1080 Ti GPUs. The STAGIN model $f$ is trained end-to-end in a supervised manner with the loss $\mathcal{L} = \mathcal{L}_{\text{xent}} + \lambda \cdot \mathcal{L}_{\text{ortho}}$

---

[2] https://db.humanconnectome.org

Table 1: Comparative study on HCP-Rest and HCP-Task dataset.

| Model | HCP-Rest | | HCP-Task | Type of FC | # Params |
| | Accuracy (%) | AUROC | Accuracy (%) | | |
| --- | --- | --- | --- | --- | --- |
| STAGIN-SERO | **88.20** $\pm$ 1.33 | **0.9296** $\pm$ 0.0187 | **99.19** $\pm$ 0.20 | Dynamic | 1,209k |
| STAGIN-GARO | 87.01 $\pm$ 3.00 | 0.9151 $\pm$ 0.0258 | **99.02** $\pm$ 0.17 | Dynamic | 1,068k |
| ST-GCN [15] | 76.95 $\pm$ 3.00 | 0.8545 $\pm$ 0.0316 | 98.92 $\pm$ 0.27 | Dynamic | 355k |
| MS-G3D [10] | 79.16 $\pm$ 2.53 | 0.8912 $\pm$ 0.0329 | - | Dynamic | 3,045k |
| BAnD++ [36] | - | - | 97.20 $\pm$ 0.57 | None | 2,010k |
| BAnD [36] | - | - | 95.10 $\pm$ 0.62 | None | 2,010k |
| r-BAnD | - | - | 98.90 $\pm$ 0.27 | Dynamic | 664k |
| GIN [23] | 81.34 $\pm$ 2.40 | 0.8955 $\pm$ 0.0237 | 93.87 $\pm$ 0.66 | Static | 169k |
| GCN [24] | 80.79 $\pm$ 2.00 | 0.8741 $\pm$ 0.0174 | 45.07 $\pm$ 1.63 | Static | 101k |
| GraphSAGE [31] | 75.48 $\pm$ 1.97 | 0.8237 $\pm$ 0.0228 | 54.52 $\pm$ 0.97 | Static | 202k |
| ChebGCN [2] | 77.76 $\pm$ 2.09 | 0.8582 $\pm$ 0.0233 | 73.06 $\pm$ 0.68 | Static | 704k |

where $\mathcal{L}_{\text{xent}}$ is the cross entropy loss and $\lambda$ is the scaling coefficient of the orthogonal regularization. We set the number of layers $K = 4$, embedding dimension $D = 128$, window length $\Gamma = 50$, window stride $S = 3$, and regularization coefficient $\lambda = 1.0 \times 10^{-5}$. The window length and stride correspond to capturing the FC within 36 seconds every 2.16 seconds, which follows the standard setting of the sliding-window dFC analyses [59, 37]. Dropout rate 0.5 is applied to the final dynamic graph representation $\boldsymbol{h}_{G_{\text{dyn}}}$, and rate 0.1 is applied to the attention vectors $\boldsymbol{z}_{\text{space}}$ and $\boldsymbol{z}_{\text{time}}$ during training. For nonlinearity $\sigma$ in (6) and (14), GELU [18] is used instead of ReLU with batch normalization before each $\sigma$. One-cycle learning rate policy is employed, which the learning rate is gradually increased from 0.0005 to 0.001 during the early 20% of the training, and gradually decreased to $5.0 \times 10^{-7}$ afterwise. Thirty training epochs were run for the HCP-Rest dataset with minibatch size 3, while ten epochs were run with minibatch size 16 for the HCP-Task dataset. We performed 5-fold stratified cross-validation of the dynamic graphs from the dataset, and report mean and standard deviation across the folds. To extract the ROI-timeseries, the Schaefer atlas [42] with 400 regions ($N = 400$) labelled with 7 intrinsic connectivity networks (ICNs) was used. The time dimension of ROI-timeseries matrix $\boldsymbol{P}$ was randomly sliced with a fixed length (600 for HCP-Rest, 150 for HCP-Task) at each steps during training for (i) relieving computational overload, (ii) stochastic augmentation of the training dataset, (iii) mitigating unwanted memorization of the specific timing of subtask onset, and (iv) matching the number of timepoints $T$ across different task labels for the HCP-Task dataset. Unsliced full matrix $\boldsymbol{P}$ was used for inference at test time. The end-to-end inference from the construction of the dynamic graph to the acquisition of the final prediction required 1.68 seconds per sample with given experimental settings.

## 5.3 HCP-Rest: Gender classification

We first validate our proposed method by gender classification on the HCP-Rest dataset. The two proposed methods, named STAGIN-GARO and STAGIN-SERO based on the type of the spatial attention module, resulted in 87.01% and 88.20% mean accuracy on the 5-fold cross validation, respectively (Table 1). The mean area under receiver operator characteristic curve (AUROC) were 0.9151 and 0.9296. Classification performance of STAGIN is compared with other GNN methods for reprensentation learning of dynamic/static FC network, including ST-GCN [15], MS-G3D [10], GIN [23], GCN [24], GraphSAGE [31], and ChebGCN [2]. We used the code by the authors of [15][3] and [10][4] but modified the cross validation scheme to avoid early stopping based on the test dataset for fair comparison. It can be seen from Table 1 that our proposed method outperforms other GNN based methods. The results of the ablation study are shown in Table 3 in the Appendix.

We use STAGIN-SERO, which showed the best accuracy, for analyzing temporal and spatial attention of the dynamic FC networks. We define the temporal attention vector $\boldsymbol{z}_{\text{time}}^{(k)} \in [0, 1]^T$ at layer $k$ as the average of row elements in the self-attention weight matrix $\boldsymbol{z}_{\text{time}}^{(k)}[j] = \frac{1}{T} \sum_{i=1}^{T} Z_{ij}$ where

---

[3] https://github.com/sgadgil6/cnslab_fmri
[4] https://github.com/metrics-lab/ST-fMRI

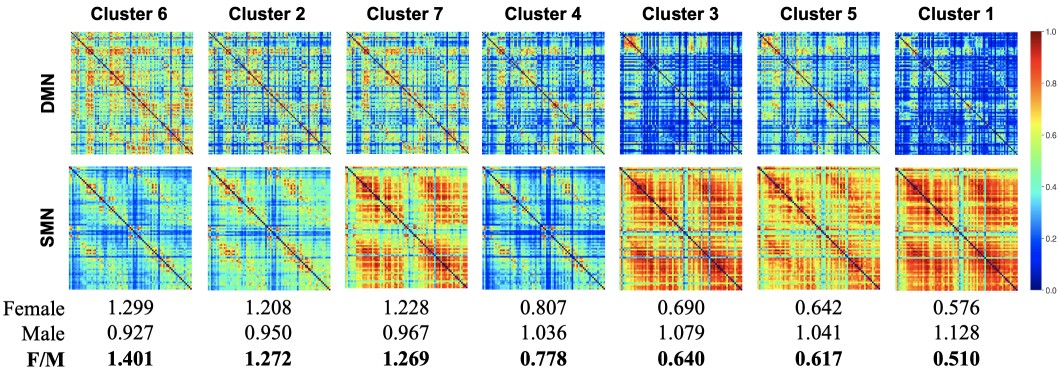

| | Cluster 6 | Cluster 2 | Cluster 7 | Cluster 4 | Cluster 3 | Cluster 5 | Cluster 1 |
|---|---|---|---|---|---|---|---|
| Female | 1.299 | 1.208 | 1.228 | 0.807 | 0.690 | 0.642 | 0.576 |
| Male | 0.927 | 0.950 | 0.967 | 1.036 | 1.079 | 1.041 | 1.128 |
| **F/M** | **1.401** | **1.272** | **1.269** | **0.778** | **0.640** | **0.617** | **0.510** |

Figure 3: Analysis of temporal attention of the gender classification experiment with k-means clustering. The DMN and SMN of the 7 cluster centroids are plotted and the relative proportion of temporally attended clusters for female and male subjects are written below. The clusters are sorted in descending order of the female/male attended cluster ratio.

$z_{\text{time}}^{(k)}[j]$ and $Z_{ij}$ are $j$-th element of $z_{\text{time}}^{(k)}$ and $(i, j)$-th element of $\mathbf{Z}_{\text{time}}^{(k)}$ for the resting-state data, respectively. To employ k-means clustering to the resting-state dynamic FC analysis [1], we first define a set of *attended* timepoints $\tilde{T} = \{t \mid z_{\text{time}}[t] > \alpha \cdot \sigma_{z_{\text{time}}}\}$ where $\alpha$ is the cutoff coefficient, and $\sigma_{z_{\text{time}}^{(k)}}$ denotes the standard deviation of $z_{\text{time}}^{(k)}$. Defining the threshold based on standard deviation inherits the practice of the point-process analysis for dynamic FC, so we set $\alpha = 1.0$ following [44]. Pattern of the FC matrices at attended timepoints $A^{\tilde{T}} = \{\mathbf{A}(t) \mid t \in \tilde{T}\}$ for each subject can now be analyzed with the k-means clustering. Specifically, we fit 7 template cluster centroids from the dynamic FC matrices $\mathbf{A}(t)$ over all subjects, and assign elements of $A^{\tilde{T}}$ into one of the 7 template clusters. The ratio of each clusters from $A^{\tilde{T}}$ with respect to $\mathbf{A}$ can then be analyzed with the subset of $A^{\tilde{T}}$ including only the female or male subjects.

Evidences from large scale studies suggest that female subjects show hyperconnectivity of the DMN [34, 39] and hypoconnectivity of the SMN when compared to male subjects [39, 13]. We accordingly hypothesized that the FC at attended timepoints will show higher values for the DMN and lower values for the SMN in female participants. Figure 3 demonstrates that the clusters mainly attended by female participants show a trend of hyperconnectivity of the DMN and hypoconnectivity of the SMN. This can be interpreted to mean that the STAGIN is properly trained to take the dynamic state of the FC networks into account for predicting the phenotype of the subject.

The spatial attention across regions of the brain is analyzed with the $z_{\text{space}}^{(k)}$ averaged across time $\tilde{z}_{\text{space}}^{(k)} := \frac{1}{T} \sum_{t=1}^{T} z_{\text{space}}^{(k)}(t)$. The regions with top 5 percentile attention values of $\tilde{z}_{\text{space}}^{(k)}$ are plotted with respect to the seven ICNs in Figure 9 in the Appendix. It can be seen that the majority of the top attended regions are from the SMN, which further suggests gender difference of resting-state FC within the SMN. A notable limitation here is that the threshold for determining the top attended region is heuristically set. Statistically determining the spatially attended regions from the resting-state data would further provide validity of the method, which is left as a future work.

## 5.4 HCP-Task: Task decoding

Task decoding refers to classifying which of the seven tasks the subject was performing during the acquisition of the brain fMRI. The STAGIN-GARO and STAGIN-SERO showed 99.02% and 99.19% mean accuracy for the task decoding experiment, respectively (Table 1). It can be seen that the proposed methods outperform the previous state-of-the-art model BAND and BAnD++ [36], which applied self-attention of the Transformer encoder directly to 3D ResNet extracted representation vectors of the fMRI without considering the network property of the brain. To account for the possible statistical disadvantage of voxel-based feature extraction, we further implemented a new region-based BAnD (r-BAnD) by using GIN without attention-based READOUT instaed of the 3D ResNet. Accuracy of r-BAnD resulted in an accuracy of 98.90%, suggesting that our method shows superior performance even when the statistical disadvantages are matched. Experiment on

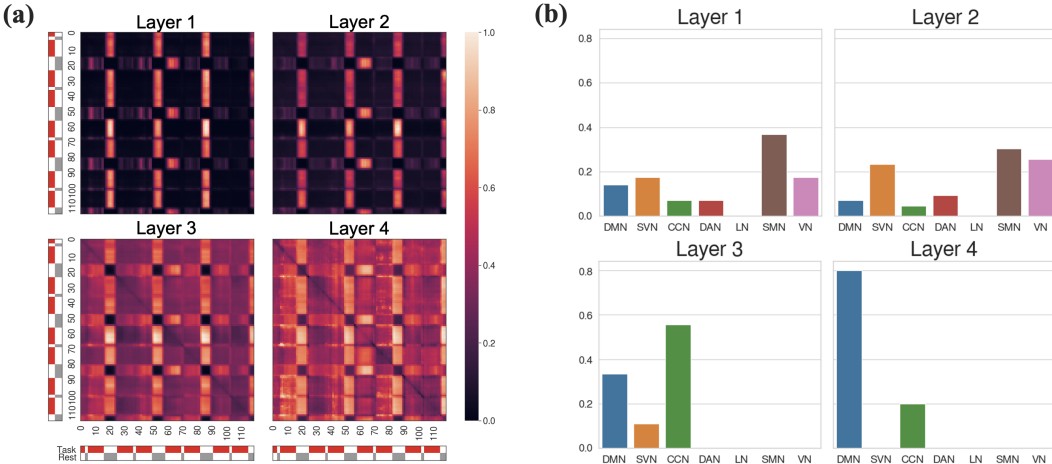

Figure 4: Analysis of spatio-temporal attention for working memory task of the task decoding experiment. (a) Plot of average temporal attention matrix $\boldsymbol{Z}_{\text{time}}^{(k)}$ across subjects. (b) Proportion of statistically significant regions within the 7 ICNs from the spatial attention GLM.

other models including ST-GCN [15], GIN [23], GCN [24], GraphSAGE [31], and ChebGCN [2] demonstrate exceptional performance of our proposed method for HCP-Task (Table 1). The fact that subtask timing information is completely lost may reflect the reason behind poor performance of static FC methods, which can be a critical disadvantage in task classification.

We interpret the result from the working memory task for spatio-temporal attention analysis, where the subtask consists of either performing an n-back memory task or rest. Our key expectation of the temporal attention analysis was that if STAGIN learns to accurately attend to temporal features of the dynamic FC graphs, then $\boldsymbol{Z}_{\text{time}}$ should represent which subtask the subject was upto. Surprisingly, it can be clearly seen that the Transformer encoder of STAGIN learns to attend to the timing of subtasks from Figure 4 (a), which demonstrates mean temporal attention $\boldsymbol{Z}_{\text{time}}$ across all subjects. Notice that no supervision is provided to the STAGIN model regarding the subtask timing during training.

To analyze the spatially attended regions $\boldsymbol{z}_{\text{space}}$ of STAGIN, we construct a GLM [14] to statistically evaluate how much each region is responsible for performing the subtasks. The parameter vectors $\boldsymbol{\beta}_{\text{task}} \in \mathbb{R}^N$ and $\boldsymbol{\beta}_{\text{rest}} \in \mathbb{R}^N$ are estimated with the sequence of spatial attention vectors and the subtask timing design matrix $M \in \{0,1\}^{T \times 2}$ by solving the following with least-squares estimation:

$$[\boldsymbol{z}_{\text{space}}(0), \cdots, \boldsymbol{z}_{\text{space}}(T)]^\top = \boldsymbol{M}[\boldsymbol{\beta}_{\text{task}}, \boldsymbol{\beta}_{\text{rest}}]^\top + \boldsymbol{\epsilon},$$

where $\boldsymbol{\epsilon}$ denotes residual error. The contrast of the estimated parameters $\hat{\boldsymbol{\beta}}_{\text{task}}$ and $\hat{\boldsymbol{\beta}}_{\text{rest}}$ was set to $\boldsymbol{c} = [1, -1]$ so the rejection of null hypothesis indicates $\hat{\boldsymbol{\beta}}_{\text{task}}[i] > \hat{\boldsymbol{\beta}}_{\text{rest}}[i]$ at the $i$-th ROI. Multiple comparisons of the $N$ ROIs are family-wise error (FWE) corrected.

Figure 4 (b) shows the proportion of statistically significant regions within the 7 ICNs for each layers. Interestingly, the layer 1 and 2 share a similar trend that the regions from SMN, visual network (VN), and salience/ventral attention network (SVN) are dominant. In contrast, layer 3 and 4 suggest a dominance of the regions from DMN and cognitive control network (CCN). We denote the layer 1 and 2 as the low-order layers (LoL) and the layer 3 and 4 as the high-order layers (HoL). The dominance of SMN and VN at LoL can be understood as the low-level sensorimotor function for perceiving the task is being processed within the short-range 1- or 2-hop connection of the networks. On the other hand, the dominance of DMN and CCN at HoL reflects the high-level cognitive integration for executing and controlling the given task being processed within the long-range 3- or 4-hop connection of the networks. Considering that the SVN is a network for integrating the low-level sensorimotor networks and the high-level executive networks to provide dynamic balancing between the two functions, the significant regions of SVN being present at both LoL and HoL is not surprising. Temporal and spatial attention plot of other six tasks are further provided in the Appendix Section D.2.

## Acknowledgments and Disclosure of Funding

This work was supported by the National Research Foundation of Korea (NRF) grant funded by the Korea government (MSIT) (No. NRF-2021M3E5D9025019, NRF-2020R1A2B5B03001980). This work was also supported by Institute of Information & communications Technology Planning & Evaluation (IITP) grant funded by the Korea government(MSIT) (No.2019-0-00075, Artificial Intelligence Graduate School Program(KAIST)) and the KAIST Key Research Institute (Interdisciplinary Research Group) Project.

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
