# A    Geometric interpretation of orthognal regularization

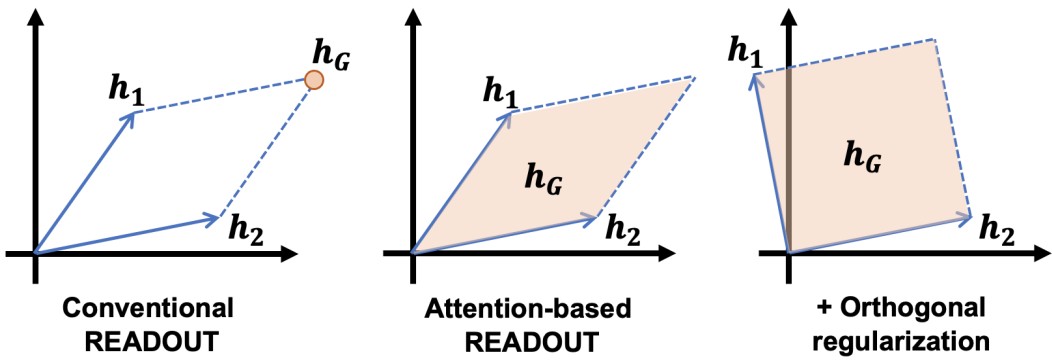

Figure 5: Geometric interpretation of the orthogonal regularization. Blue arrows indicate node feature vectors $h_v$ of the latent space, and the orange area/point indicate possible range of graph feature vector $h_G$ obtained by applying READOUT to $h_v$.

We elaborate our motivation behind orthogonal regularization (15) proposed in Section 4.2.3. The biggest motivation behind orthognoal regularization lies in understanding (8) and (12) that the node features $H$ becomes full rank matrix with good condition number. Figure 5 visually demonstrates the geometric effect of attention-based READOUT and orthogonal regularization with two example node features $h_1$ and $h_2$. Only one graph feature vector $h_G$ is possible from the combination of two node features with conventional READOUT, while vectors within the range of the orange rhombus can represent the whole graph feature with attention-based READOUT. With orthogonal regularization, area of the range that the graph feature vector $h_G$ can represent become even larger, with lower possibility of null subspace within $H$. Accordingly, the subspace that $H$ can span can be rich enough.

# B    Detailed description of the dataset

Detailed description of the experiment datasets are summarized in Table 2. Baseline subtask for serving as the control condition, such as *Rest* or *Response*, are listed as the last item in the Subtasks column. An important fact about the HCP is that a large number data from twin subjects are included within the dataset. While this fact has been largely ignored in previous GNN-fMRI studies of gender classification using the HCP dataset, biological influence of shared genetic background on the FC can be quite significant. We did not take this into account in this work to make a more straightforward comparison with previous methods, but it should be noted as a limitation of this research that requires careful consideration in related future studies.

Table 2: Description of the experiment datasets.

| Dataset | Task type | Subtasks | No. images | $T_{\max}$ | $C$ |
|---------|-----------|----------|------------|------------|-----|
| HCP-Rest | Resting-state | Rest | 1093 | 1200 | 2 |
| HCP-Task | Working Memory | Task, Rest | 1087 | 405 | |
| | Social | Mental, Random, Rest | 1053 | 274 | |
| | Relational | Task, Rest | 1043 | 232 | |
| | Motor | (L,R).(Hand,Foot), Tongue, Rest | 1085 | 284 | 7 |
| | Language | Story, Math, Response | 1051 | 316 | |
| | Gambling | Task, Rest | 1082 | 253 | |
| | Emotion | Shape, Face, Rest | 1049 | 176 | |

# C Additional experiment results

## C.1 Ablation study

Ablation study results are provided in Table 3. The results suggest that STAGIN shows degraded performance by ablating the orthogonal regularization ($\mathcal{L}_{\text{ortho}}$), spatial attention ($z_{\text{space}}$), temporal attention ($Z_{\text{time}}$), and timestamp encoding $\eta(t)$, confirming the importance of each components of the model. Gain of classification performance by applying spatio-temporal attention is not as significant as by applying timestamp encoding, but the attention modules are still uncompensable in that they provide neuroscientific explainability of the model. Extracting the ROI-Timeseries matrix $P$ with other widely used atlases including AAL, Destrieux, and Harvard-oxford are also experimented, and confirmed that the Schaefer atlas with 400 ROIs show best classification performance.

Table 3: Ablation study results.

| Atlas | $N$ | $\mathcal{L}_{\text{ortho}}$ | $z_{\text{space}}$ | $Z_{\text{time}}$ | $\eta(t)$ | Accuracy (%) | AUROC |
|---|---|---|---|---|---|---|---|
| | | ✓ | ✓ | ✓ | ✓ | $88.20 \pm 1.33$ | $0.9296 \pm 0.0187$ |
| | | ✗ | ✓ | ✓ | ✓ | $87.46 \pm 3.56$ | $0.9213 \pm 0.0242$ |
| Schaefer | 400 | ✗ | ✗ | ✓ | ✓ | $86.55 \pm 3.12$ | $0.9260 \pm 0.0216$ |
| | | ✗ | ✗ | ✗ | ✓ | $85.64 \pm 2.47$ | $0.9272 \pm 0.0104$ |
| | | ✗ | ✗ | ✗ | ✗ | $82.34 \pm 3.38$ | $0.9005 \pm 0.0256$ |
| AAL | 116 | ✓ | ✓ | ✓ | ✓ | $85.36 \pm 1.58$ | $0.9216 \pm 0.0116$ |
| Destrieux | 150 | ✓ | ✓ | ✓ | ✓ | $85.73 \pm 1.39$ | $0.9235 \pm 0.0126$ |
| Harvard-oxford | 48 | ✓ | ✓ | ✓ | ✓ | $82.07 \pm 1.11$ | $0.9008 \pm 0.0093$ |

## C.2 Hyperparameter experiments

Hyperparameter experiment results are provided in Table 4. The model tends to be robust to hyperparameter changes, and showed even better HCP-Rest gender classification performance when the edge threshold was set to 40% instead of 30% (bold numbers in Table 4).

Table 4: Hyperparameter experiment results.

| Hyperparameter | | Accuracy (%) | AUROC |
|---|---|---|---|
| | 20% | $88.01 \pm 2.81$ | $0.9304 \pm 0.0220$ |
| Edge threshold | *30% | $88.20 \pm 1.33$ | $0.9296 \pm 0.0187$ |
| | 40% | $\mathbf{89.02 \pm 1.80}$ | $\mathbf{0.9408 \pm 0.0110}$ |
| | 25 (18s) | $85.45 \pm 3.51$ | $0.9252 \pm 0.0235$ |
| $\Gamma$ | * 50 (36s) | $88.20 \pm 1.33$ | $0.9296 \pm 0.0187$ |
| | 75 (54s) | $86.37 \pm 1.87$ | $0.9218 \pm 0.0168$ |
| | $1.0 \times 10^{-4}$ | $87.46 \pm 2.56$ | $0.9336 \pm 0.0179$ |
| $\lambda$ | *$1.0 \times 10^{-5}$ | $88.20 \pm 1.33$ | $0.9296 \pm 0.0187$ |
| | $1.0 \times 10^{-6}$ | $88.10 \pm 2.08$ | $0.9347 \pm 0.0194$ |

\* Asterisks indicate baseline experiment settings

## C.3 Comparative experiment of spatial attention scoring

While the motivation may have been different, our attention-based READOUT functions share methodological similarity with graph pooling methods, which score and rank each nodes within the graph for the selection of important nodes. We experimented on replacing our attention-based READOUT functions with some well-known graph pooling methods including TopKPooling [16], SAGPooling [28], ASAPooling [38] from the PyTorch Geometric[5] package [12] without dropping any vertices for scoring the level of attention across the nodes. The results suggest that our attention-based

---

[5]https://pytorch-geometric.readthedocs.io/

READOUT functions perform better and more stable, with lower computational overload for our graph classification task.

Table 5: Comparison with pooling methods for scoring spatial attention.

| Module | Accuracy (%) | AUROC |
|---|---|---|
| SERO (Ours) | $88.20 \pm 1.33$ | $0.9296 \pm 0.0187$ |
| TopKPooling [16] | $77.02 \pm 10.94$ | $0.8203 \pm 0.1123$ |
| SAGPooling [28] | OOM | OOM |
| ASAPooling [38] | OOM | OOM |

We believe that the strength of our attention-based READOUT comes from taking the globally pooled graph feature $H\Phi_{\text{mean}}$ as a prior, which may represent the whole graph property better than a randomly initialized learnable vector (TopKPooling) or GNN aggregated close neighborhood information (SAGPooling, ASAPooling).

# D    Additional attention analysis results

## D.1    Temporal attention of HCP-Rest

Analysis of the HCP-Rest temporal attention of are further analyzed with (i) varying number of clusters for k-means clustering, (ii) comparing with unattended average FC pattern in female and male subjects, and (iii) statistical testing of cluster-by-gender attending frequency.

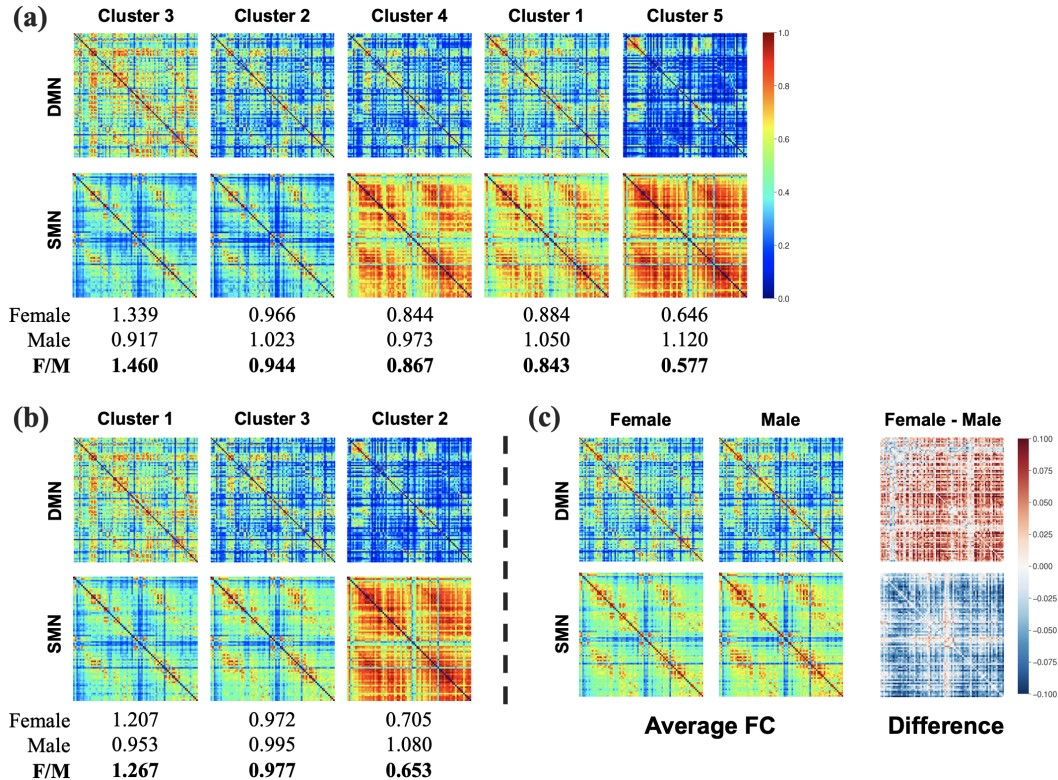

Figure 6: Clustering analysis of HCP-Rest temporally attended regions with (a) number of clusters set to 5, and (b) number of clusters set to 3. (c) Plot of the unattended average FC matrix for female and male subjects. Female subjects show slight hyperconnectivity in the DMN and hypoconnectivity in the SMN when compared to male subjects.

Figure 6 (a) and (b) demonstrate the clustering analysis result with number of cluster centroids set to 5 and 3, respectively. It can be seen that the same pattern of DMN hyperconnectivity and SMN hypoconnectivity is found irrespective of the number of clusters. Figure 6 (c) show a plot of average DMN and SMN connectivity in female and male subjects, which have minimal difference between the two genders. When the difference is computed by subtracting average FC matrix of female subjects by that of male subjects, a slight hyperconnectivity in DMN and

Table 6: Chi-square test of temporal attending frequency

| Layer | $\chi^2$ | $p$ |
| --- | --- | --- |
| 1 | 668.583 | <0.001 |
| 2 | 649.589 | <0.001 |
| 3 | 433.615 | <0.001 |
| 4 | 420.542 | <0.001 |

hypoconnectivity in SMN is present in the average pattern. This average pattern again confirms the validity of our method by showing that our method can capture the small difference between the two groups that is present in the dynamic FC graph, and exploit the captured information for classification. Chi-square test on the difference of attending frequency between the cluster-by-gender resulted in that the frequency of attended clusters are significantly different between female and male subjects (Table 6).

## D.2 Temporal and spatial attention analysis of all task types from HCP-Task

Temporal (Figure 7) and spatial (Figure 8) attention analysis results of task types other than working memory are provided in this section. It can be seen from Figure 7 that the Transformer encoder of STAGIN learns to temporally attend to the subtasks regardless of the task type, without any subtask timing information provided during training.

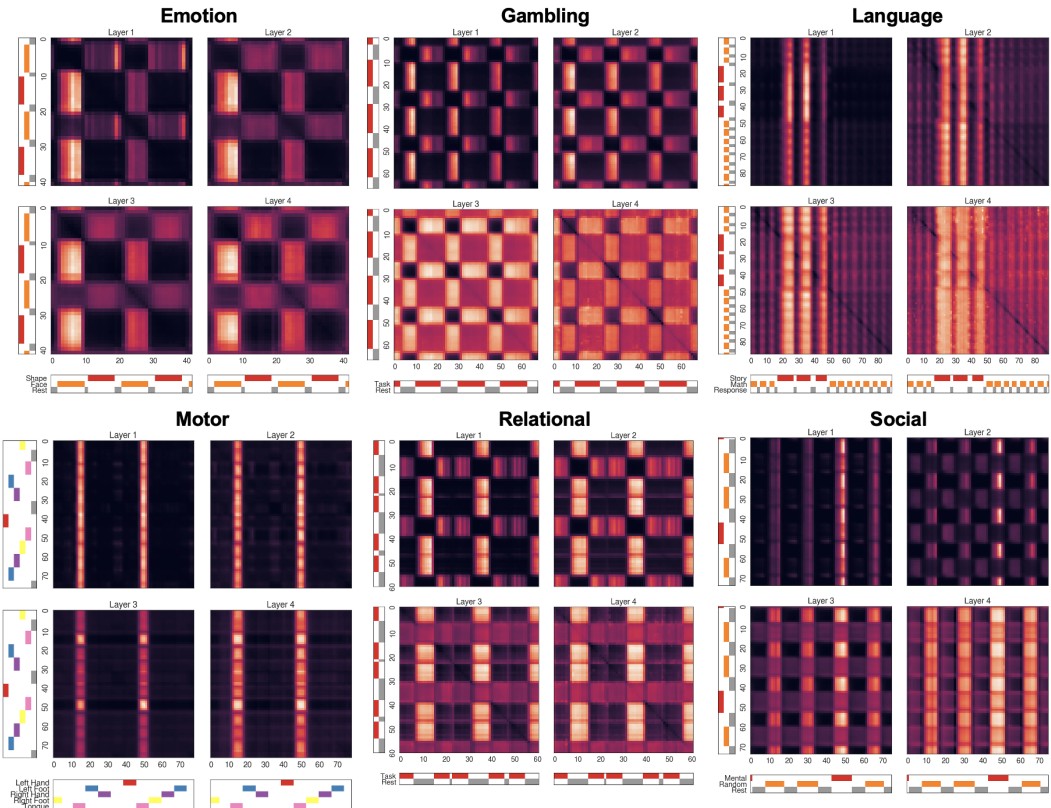

Figure 7: Plot of the HCP-Task temporal attention $Z_{\text{time}}^{(k)}$ averaged across all subjects.

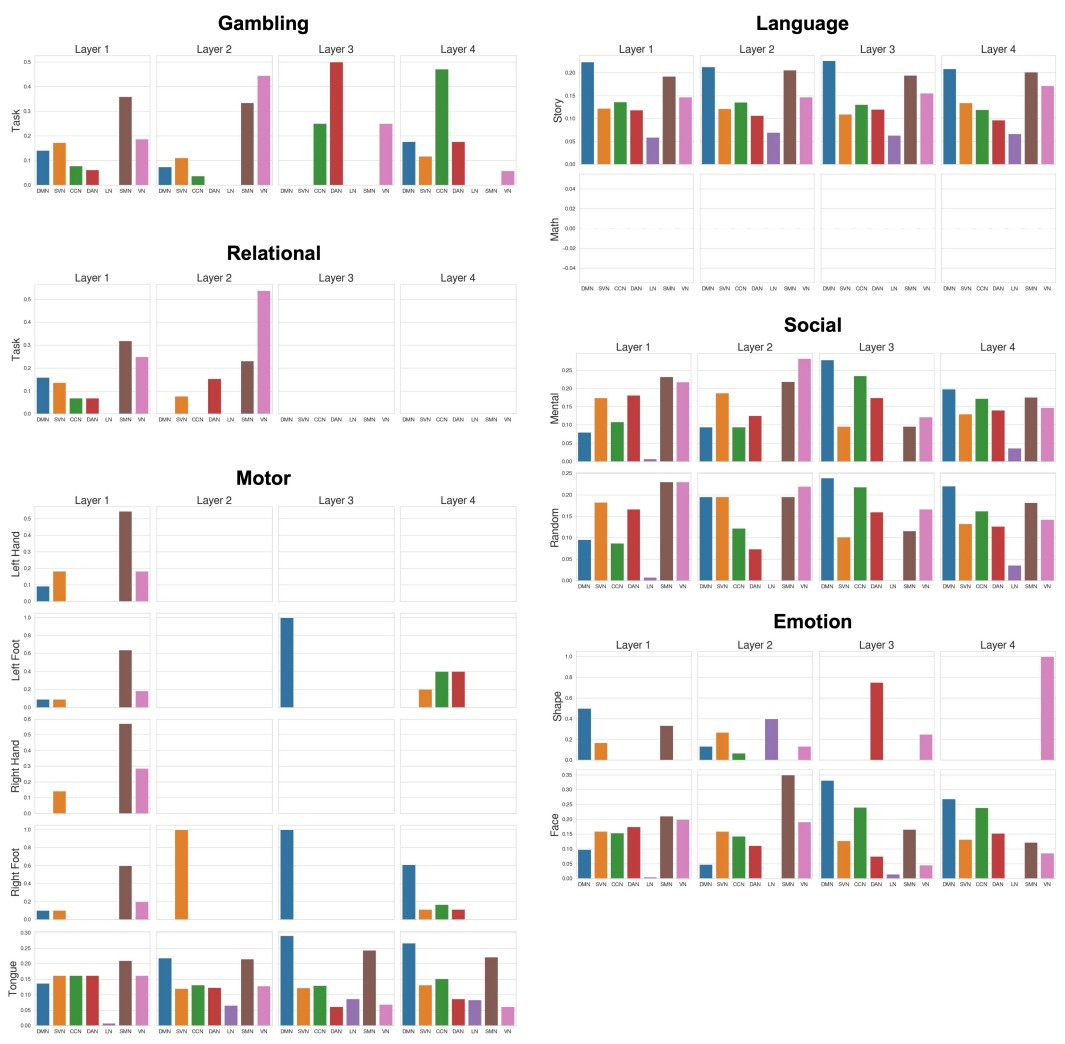

Figure 8: Proportion of statistically significant regions within the 7 ICNs from the HCP-Task spatial attention GLM. Each subtask is contrasted with the baseline subtask, i.e. rest or response.

# E    Brain plot of spatially attended regions from HCP-Rest and HCP-Task

Spatially attended regions of the HCP-Rest and HCP-Task experiments are visualized on a template brain with respect to the 7 ICNs and the four STAGIN layers in Figure 9 and 10. Ratio of significant regions between the two hemispheres and the 7 ICNs are also demonstrated as pie plots. Defining the spatially attended regions follow the result of GLM statistical significance ($p$-FWE $< 0.05$) for the HCP-Task, and the regions with top 5-percentile attention score for the HCP-Rest.

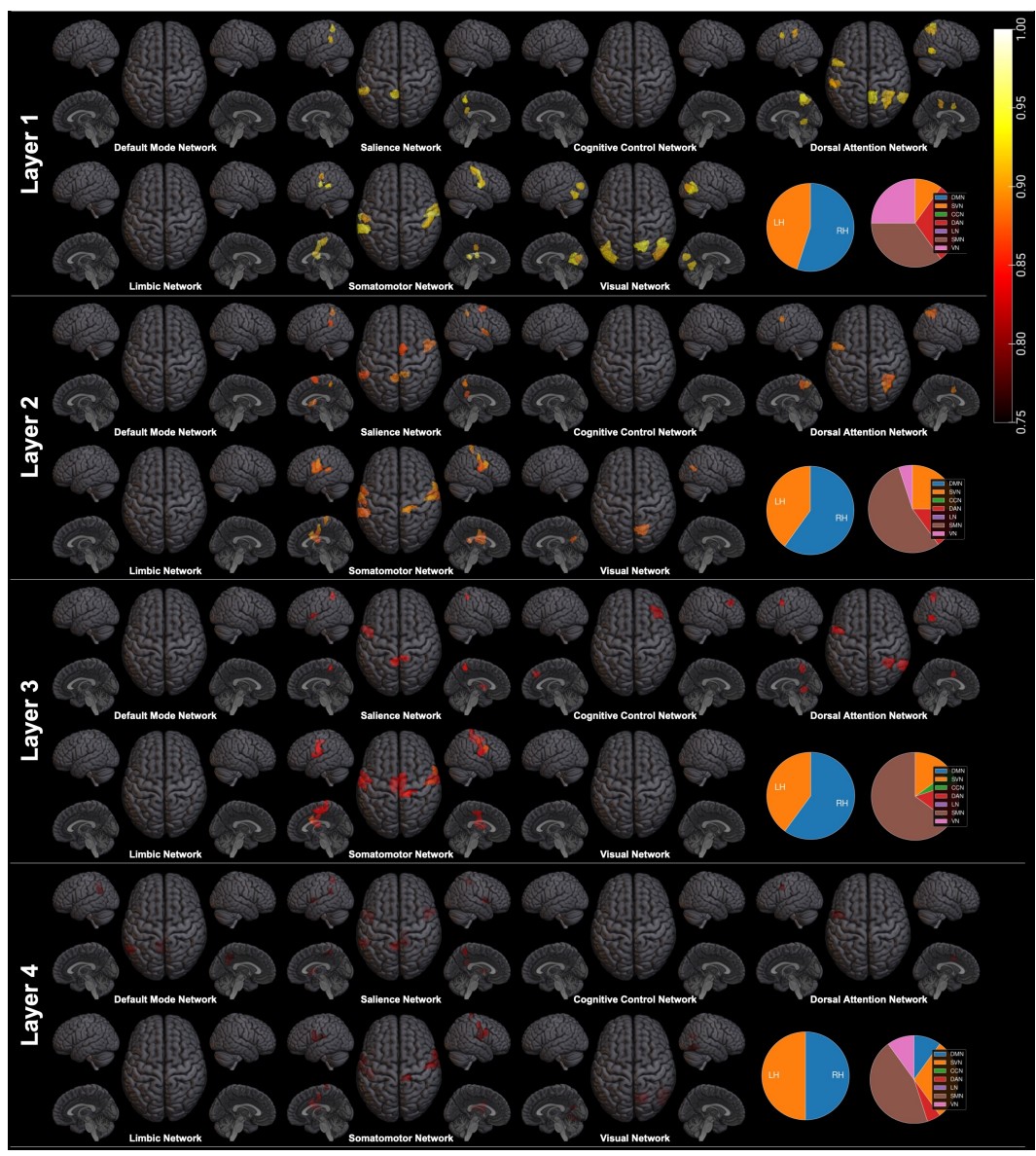

Figure 9: Brain plot of top 5-percentile HCP-Rest spatial attention regions.

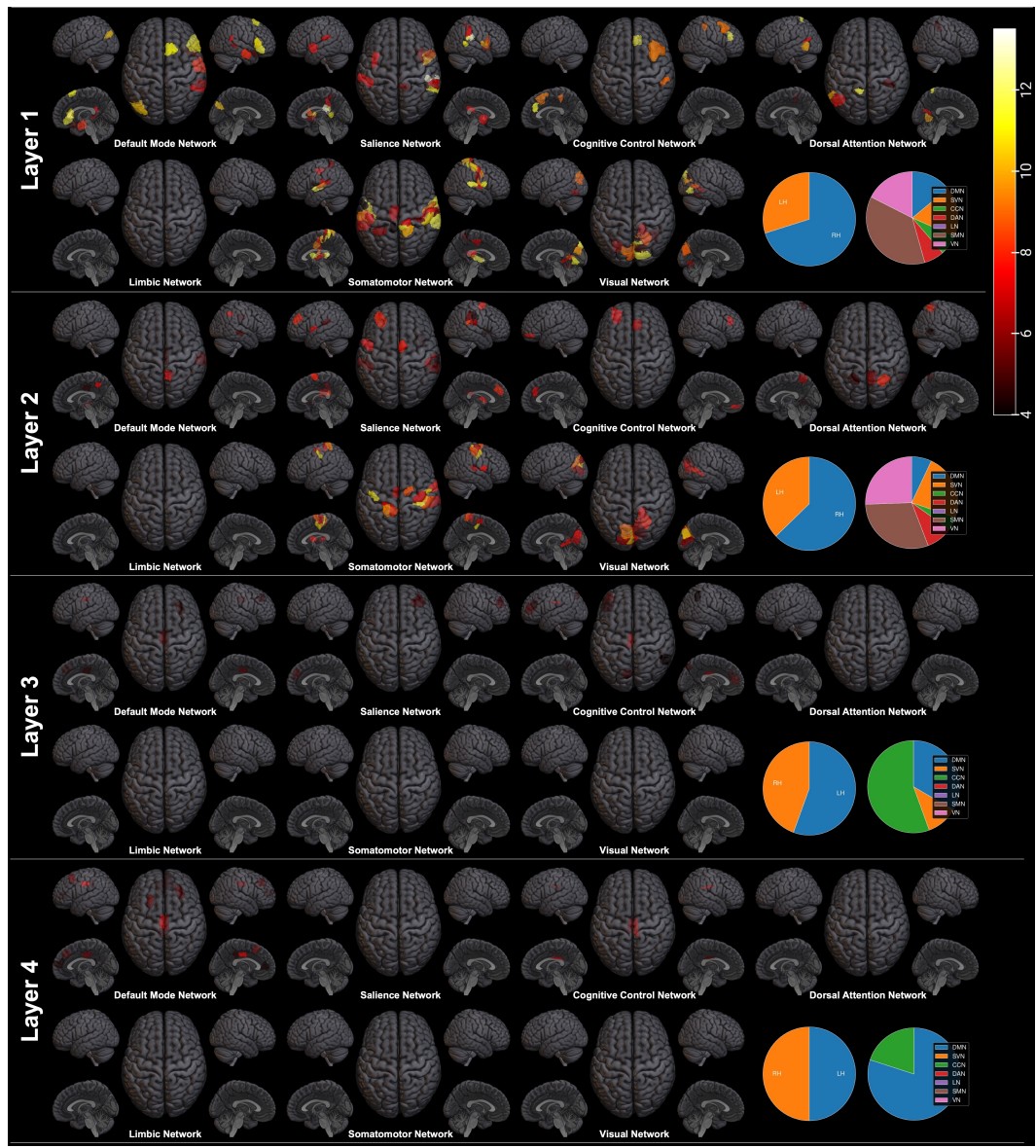

Figure 10: Brain plot of statistically significant HCP-Task working memory spatial attention regions.