# OpenReview forum: "Learning Dynamic Graph Representation of Brain Connectome with Spatio-Temporal Attention"
_NeurIPS.cc/2021/Conference — NeurIPS 2021 Poster_

### Official Review · Reviewer_i7Br · 2021-07-14

**Rating:** 7
**Confidence:** 4

**Summary:**

This paper proposes a method, STAGIN, to model dynamic connectivity. The architecture combines graph isomorphic network (GIN) with spatial and temporal attention. STAGIN is applied to HCP data to classify gender as well as tasks with higher accuracy than state-of-the-art methods shown.

**Ethical Concerns:**

There are no ethical concerns in this paper.

**Limitations And Societal Impact:**

Description of limitations and negative societal impact is sparse. Might be helpful to mention some of the limitations stated above especially on result interpretation and the chosen experiments.

**Main Review:**

Originality
While each component of STAGIN is based on existing works, the proposed combination shows improved performance over state-of-the-art.

Quality
The method seems sounded, but some claims and results raise concerns.
1. It is mentioned that “we address the issue that the node features of the input dynamic graph do not contain any temporal information and concatenate encoded timestamp with the node features”, but [11] uses windowed sub-sequences as node features of ST-GCN, which is analogous to the timestamp used in STAGIN, with the difference being not all sub-sequences are fed in simultaneously. Also, [3] uses temporal convolution network at the input, which would capture temporal dynamics. This latter method should be quantitatively compared and not brush off with one sentence.
2. Temporal explainability is claimed to be a limitation of existing methods. For resting state fMRI, why would someone care which time points are most relevant for classifying gender or other phenotypes. For task fMRI, while it is reassuring to see attention is paid at the time of stimulus, are we gaining any new information? Please provide examples of when temporal explainabiltiy is useful.
3. For interpretation, “k-means clustering analysis of the resting-state dynamic FC [1] and general linear model (GLM) statistical mapping of task fMRI [10] for interpreting the spatio-temporal attention learned from STAGIN” were used. The question is why can’t we apply these to latent representations or other models? This is not a particular property of STAGIN.
4. How sensitive are results to thresholding the top 30-percentile values of the correlation matrix? How were other hyper-parameters chosen?
5. How were subjects divided into folds? HCP is a twin study hence subjects are correlated. Were one of the twins removed for each pair?
6. How was temporal correlation accounted for when dividing the timeseries in the case of task classification? Or was classification performed at subject level with entire session timeseries taken as one sample, in which case, how was correlation between sessions from the same subject accounted for?
7. Would like to see STAGIN applied to harder tasks e.g. ASD classification.
8. The current experiments and the way results are visually presented (e.g. Figures 3, 6, and 7) do not provide much neuroscience insights. Would be nice to see less superficial neuroscience findings to show the gain in interpretability by STAGIN.

Clarity
Most parts of the paper are clearly written but technical details/contributions are deferred to references, partly since the proposed architecture is mainly based on existing works. A few minor points that might be worth considering below.
1. While it might be true that this is the “first study to demonstrate the capability to learn the dynamic graph representation of both resting-state and task fMRI data with a single framework”, why is this difficult? Any general DL architecture can be applied to both rest and task fMRI data.
2. The clear description of GIN is appreciated and the “encoder/decoder understanding of GIN” does provide some motivations to use attention instead of sum/average. However, that section seems to be an overkill to say attention is better than just sum/average.
3. Figure 3 is hard to follow. For example, cluster 7 is supposedly attended more by females but SMN has bright red blocks, i.e. not hypoconnectivity. The qualitative evaluation needs more work. Is the ratio of female/male attentiveness significantly different from 1 statistically?

Significance
It is a reasonable empirical study showing that graph convolution followed by spatial and temporal attention can improve fMRI classification. The current classification task might not be difficult enough to show the true gain of STAGIN. Could potentially be of broad interest if the result interpretation part is further developed.

Post Author Reponses
I have read through the reviews and responses, and will keep my original score since that score assumed correct statistical setup for the classification evaluation, which the authors have clarified and "promised" to acknowledge the issue with twin data. A point worth noting is that just finding which time points drive classification does not help interpretation for resting state data. What might help is the relative number of time points with "aberrant signals" between subject groups for example.

**Time Spent Reviewing:**

6

---

> ### Author Response · Authors · 2021-08-10
> **Response to reviewer i7Br [2/2]**
>
>
>
> ### Clarity
> > 1. While it might be true that this is the “first study to demonstrate the capability to learn the dynamic graph representation of both resting-state and task fMRI data with a single framework”, why is this difficult? Any general DL architecture can be applied to both rest and task fMRI data.
>
>
> We agree with the reviewer's comment and will clarify that our paper is the first, but not only possible application to both resting-state and task fMRI data on the revised manuscript.
>
>
> > 2. The clear description of GIN is appreciated and the “encoder/decoder understanding of GIN” does provide some motivations to use attention instead of sum/average. However, that section seems to be an overkill to say attention is better than just sum/average.
>
> The description will be down-toned on the revised manuscript.
>
>
> > 3. Figure 3 is hard to follow. For example, cluster 7 is supposedly attended more by females but SMN has bright red blocks, i.e. not hypoconnectivity. The qualitative evaluation needs more work. Is the ratio of female/male attentiveness significantly different from 1 statistically?
>
> We apologize for the unclear demonstration.
> The Figure 3. will be more clearly demonstrated in the revised manuscript, as mentioned on the above response to the comment 8.
>
> The cluster 7 does seem to not follow the trend for the SMN.
> We expect that this is related to the number of clusters being unnecessarily large, generating a redundant cluster centroid with hyperconnectivity in both DMN and SMN.
> The number of cluster was originally set to follow a previous work by [3], which we find this too large for our interpretation and plan to report clusters with 5 and 3 centroids.
> We find that reducing the number of clusters demonstrate a more precise interpretation of our method.
>
> Chi-square test on the difference of attending frequency between the cluster-by-gender was performed and resulted in that the frequency of attended clusters are different between female and male subjects:
>
> | Layer | $\chi^{2}$ | p-value |
> |---|---|---|
> |   1   |  668.583   | < 0.001 |
> |   2   |  649.589   | < 0.001 |
> |   3   |  433.615   | < 0.001 |
> |   4   |  420.542   | < 0.001 |
>
>
> [3] Allen, Elena A., et al. "Tracking whole-brain connectivity dynamics in the resting state." Cerebral cortex 24.3 (2014): 663-676.
>
> ## Limitations And Societal Impact
> > Description of limitations and negative societal impact is sparse. Might be helpful to mention some of the limitations stated above especially on result interpretation and the chosen experiments.
>
> We believe that all attempts to decode trait and state from human brain signal measurements have potential negative societal impact of abuse or misuse.
> For example, a method that can accurately decode what someone is thinking from fMRI signals can raise privacy concerns.
> Although our method is yet far behind the decoding capability that can be abused or misused, our research cannot still be free from these ethical considerations.
> This negative societal impact will be further elaborated in the revised manuscript.

---

> ### Author Response · Authors · 2021-08-10
> **Response to reviewer i7Br [1/2]**
>
> ## Main Review
> ### Quality
> > 1. It is mentioned that “we address the issue that the node features of the input dynamic graph do not contain any temporal information and concatenate encoded timestamp with the node features”, but [11] uses windowed sub-sequences as node features of ST-GCN, which is analogous to the timestamp used in STAGIN, with the difference being not all sub-sequences are fed in simultaneously. Also, [3] uses temporal convolution network at the input, which would capture temporal dynamics. This latter method should be quantitatively compared and not brush off with one sentence.
>
> Thank you for the important point!
> We agree with the comment that the sentence can be misleading, and will be corrected in the revised manuscript.
> Also, the TCN-GN-DiffPool from [3] will be quantitatively compared with our method which is currently under re-implementation.
> For a quick comparison, we mention that the reported accuracy (%) and AUROC of TCN-GN-DiffPool are $76 \pm 1.0$ and $0.84 \pm 0.009$, respectively.
>
>
>
> > 2. Temporal explainability is claimed to be a limitation of existing methods. For resting state fMRI, why would someone care which time points are most relevant for classifying gender or other phenotypes. For task fMRI, while it is reassuring to see attention is paid at the time of stimulus, are we gaining any new information? Please provide examples of when temporal explainabiltiy is useful.
>
>
> Understanding the underlying meaning of temporal variation of dynamic FC at resting-state has been one of the key interests of neuroimaging researchers [1,2].
> This interest is based on findings that support the hypothesis that aberrant signals may be present at certain timing of the resting-state dynamic FC for particular neuro-psychiatric illnesses, which cannot be captured by the static resting-state FC.
> Finding timepoints that are relevant to classifying certain phenotypic difference based on fMRI is expected to help reveal this aberrant temporal signal, which can serve as a biomarker for the phenotypic difference of individuals in the future.
> On the other hand, not much more than validating our method is expected from interpreting the temporal attention task-fMRI data, where confirming the validity of temporal attention was still important in our research which first tried to propose a plausible temporal explainability method of the dynamic FC.
>
> [1] Hutchison, R. Matthew, et al. "Dynamic functional connectivity: promise, issues, and interpretations." Neuroimage 80 (2013): 360-378.\
> [2] Preti, Maria Giulia, Thomas AW Bolton, and Dimitri Van De Ville. "The dynamic functional connectome: State-of-the-art and perspectives." Neuroimage 160 (2017): 41-54.
>
> > 3. For interpretation, “k-means clustering analysis of the resting-state dynamic FC [1] and general linear model (GLM) statistical mapping of task fMRI [10] for interpreting the spatio-temporal attention learned from STAGIN” were used. The question is why can’t we apply these to latent representations or other models? This is not a particular property of STAGIN.
>
>
> The two analyses can indeed be generally applied to other methods as long as it provides spatial and temporal explainability similar to work.
> We will revise the manuscript that this can be made more clear in the main text.
> As the reviewer pointed out, the GLM and the k-means clustering are actually widely accepted methods for interpreting task and dynamic resting-state fMRI data, respectively.
> The motivation for interpreting attention with these two analyses also comes from this fact, which we believe that analyses results replicating previous neuroimaging knowledge with widely accepted methods can better validate our idea.
> We had no intent to put emphasis on the GLM and k-means clusterings as a particular property of STAGIN, and will further revise the manuscript to make no confusion to the readers.
>
>
> > 4. How sensitive are results to thresholding the top 30-percentile values of the correlation matrix? How were other hyper-parameters chosen?
>
> Thank you very much for the important comment.
> We find that the proposed method tend to be robust to hyperparameter changes, including the edge threshold.
>
> | Edge threshold |   Accuracy   |      AUROC      |
> |---|---|---|
> |   20%    | 88.01 (2.81) | 0.9304 (0.0220) |
> |   *30%   | 88.20 (1.33) | 0.9296 (0.0187) |
> |   40%    | **89.02** (1.80) | **0.9408** (0.0110) |
>
> It should be noted that the edge threshold 40% resulted in better accuracy and AUROC (bold numbers) than the reported metric from our submission, which will be discussed in the updated manuscript.
> We appreciate the suggestion from the reviewer, which helped us improve our method further!
>
> Other hyperparameter experiments are performed and they further support robustness of our method.
>
> |  $\Gamma$   |   Accuracy   |      AUROC      |
> |---|---|---|
> | 25 (18s) | 85.45 (3.51) | 0.9252 (0.0235) |
> | *50 (36s)| 88.20 (1.33) | 0.9296 (0.0187) |
> | 75 (54s) | 86.37 (1.87) | 0.9218 (0.0168) |
>
>
> |   $\lambda$    |   Accuracy   |      AUROC      |
> |---|---|---|
> | $1.0 \times 10^{-4}$ | 87.46 (2.56) | 0.9336 (0.0179) |
> | *$1.0 \times 10^{-5}$ | 88.20 (1.33) | 0.9296 (0.0187) |
> | $1.0 \times 10^{-6}$ | 88.10 (2.08) | 0.9347 (0.0194) |
>
>
> > 5. How were subjects divided into folds? HCP is a twin study hence subjects are correlated. Were one of the twins removed for each pair?
>
> The subjects were divided into each folds by stratified k-fold cross validation scheme.
> Thus, class imbalance was taken into account when dividing the subjects, but not the existence of twin subjects.
> We see that the point raised by the reviewer can be quite significant, but has been largely ignored in GNN-fMRI studies of gender classification using the HCP dataset.
> Although a more straightforward comparison of classification performance could be made with previous methods which also did not take the existence of twin subjects into consideration, we will discuss this as a limitation and ask for future works to consider this point in the revised manuscript.
>
>
> > 6. How was temporal correlation accounted for when dividing the timeseries in the case of task classification? Or was classification performed at subject level with entire session timeseries taken as one sample, in which case, how was correlation between sessions from the same subject accounted for?
>
>
> To mitigate the unwanted memorization of the specific timing of the task onset during training the model, we sliced the full ROI-timeseries matrix into a matrix with 150 timepoints with randomly chosen initial point (line 220 of the paper).
> This ensured that the model to be trained with random task onset timepoints even if the samples were from the same task.
> We appreciate the reviewer for clarifying this point, which will be mentioned as another (4th) motivation for the random slicing of the training input (line 221-223 of the paper) in the revised paper.
>
>
> > 7. Would like to see STAGIN applied to harder tasks e.g. ASD classification.
>
> Application of STAGIN to clinical data is planned to follow this work, which was beyond the scope of this paper.
> To tackle the difficulties that come with fMRI data in clinical samples (lack of large-scale data, variation of image quality across institutions, lower resolution and shorter timepoints, etc.), we are working on self-supervised pre-training of STAGIN with large scale resting-state dataset from healthy subjects, and fine-tuning the model on clinical samples.
>
>
> > 8. The current experiments and the way results are visually presented (e.g. Figures 3, 6, and 7) do not provide much neuroscience insights. Would be nice to see less superficial neuroscience findings to show the gain in interpretability by STAGIN.
>
>
> For Figure 3. our motivation was to demonstrates the validity of our method by replicating prior neuroimaging evidences on gender difference, rather than providing a new insight.
> We plan to improve the interpretability of Figure 3 by adding:
> i) Other number of clusters for k-means clustering will be demonstrated (k=5 and k=3) which we confirmed to have same trend with k=7.
> ii) Average functional connectivity matrix of unattended regions of female and male subjects for baseline comparison.
> iii) Chi-square statistical test results for quantitative comparison.
>
> The Figures 6 and 7 are shown to provide a quick glimpse of how the learned temporal and spatial attention look like.
> We will consider removing the Figures given that it can be too much, and misleading to the readers.

---

### Official Review · Reviewer_qvpg · 2021-07-15

**Rating:** 3
**Confidence:** 5

**Summary:**

The authors propose the Spatio-Temporal Attention Graph Isomorphism Network (STAGIN),  which can learn the dynamic graph representation of the brain connectome with spatio-temporal attention. The paper uses visualization for the interpretable display to a certain extent.

**Limitations And Societal Impact:**

Limitations
-Simple combination of existing technologies
-The performance of the proposed model is affected by hyperparameters.
-Interpretability

**Main Review:**

The paper proposes a new dynamic FC graph model and explores the interpretability of spatio-temporal attention. This is an interesting research work. But there are still some issues that need to be clarified.

1、The innovation of the paper is limited. Both the attention mechanism and transformer are existing technologies. Authors do not highlight their proposed technology details and technical novelty. In addition, the paper lacks theoretical innovation.

2、The authors claim that the proposed method achieves SOTA performance. The authors introduced a lot of related work in the Introduction, but only used 2 baseline methods. The authors should increase the baseline methods and report the method parameters and running time, etc.

3、The effect of hyperparameters on the results is not discussed. For example, λ and edge threshold (30%).

4、There are many methods to score attention. Why did the authors only choose SERO and GARO？

5、The author did not perform an interpretable analysis of the constructed dynamic graph (the graph before spatio-temporal attention). This is also important for verifying the authors' research motivation or interpretability.

6、More description should be added to the caption in Figure 1.

7、Figure 2 does not seem to be a complete display. For example, there is no node feature $\boldsymbol{x}_{v}(t)$.



**Time Spent Reviewing:**

8

---

> ### Author Response · Authors · 2021-08-10
> **Response to reviewer qvpg [2/2]**
>
>
> > 4. There are many methods to score attention. Why did the authors only choose SERO and GARO？
>
> While the motivation may have been different, we agree with the reviewer that our attention-based READOUT functions share methodological similarity with graph pooling methods, which score and rank each nodes within the graph for the selection of important nodes.
> We experimented on replacing our attention-based READOUT functions with some well-known graph pooling methods ([TopKPooling](https://pytorch-geometric.readthedocs.io/en/latest/modules/nn.html#torch_geometric.nn.pool.TopKPooling), [SAGPooling](https://pytorch-geometric.readthedocs.io/en/latest/modules/nn.html#torch_geometric.nn.pool.SAGPooling), [ASAPooling](https://pytorch-geometric.readthedocs.io/en/latest/modules/nn.html#torch_geometric.nn.pool.ASAPooling) from the [PyTorch Geometric](https://pytorch-geometric.readthedocs.io/en/latest/index.html) package) without dropping any vertices for scoring the level of attention across the nodes.
> The results suggest that our attention-based READOUT functions perform better and more stable, with lower computational overload for our graph classification task.
>
> | READOUT |   Accuracy   |      AUROC      |
> |---|---|---|
> |   SERO  | 88.01 (2.81) | 0.9304 (0.0220) |
> |   TopKPooling  | 77.02 (10.94) | 0.8203 (0.1123) |
> |   SAGPooling   | OOM | OOM |
> |   ASAPooling   | OOM | OOM |
>
> We believe that the strength of our attention-based READOUT comes from taking the globally pooled graph feature ($\mathbf{H} \Phi_{\text{mean}}$) as a prior, which may represent the whole graph property better than a randomly initialized learnable vector (TopKPooling) or GNN aggregated close neighborhood information (SAGPooling, ASAPooling).
> Extending our attention-based READOUT functions for graph pooling can be an interesting future study, which was beyond the scope of this research.
>
>
> > 5. The author did not perform an interpretable analysis of the constructed dynamic graph (the graph before spatio-temporal attention). This is also important for verifying the authors' research motivation or interpretability.
>
>
> The motivation for constructing dynamic FC graph inherits from the large interest in the topic of the neuroimaging community [13,14].
> We plan to provide qualitative analysis of the constructed graph by visual demonstration of the temporal change of the connectivity as the reviewer suggested.
> Average functional connectivity of female and male subjects within DMN and SMN will also be provided along with Figure 3 for highlighting the attended timepoints with respect to the graph before attention to improve interpretability of our temporal attention method.
>
> [13] Hutchison, R. Matthew, et al. "Dynamic functional connectivity: promise, issues, and interpretations." Neuroimage 80 (2013): 360-378.\
> [14] Preti, Maria Giulia, Thomas AW Bolton, and Dimitri Van De Ville. "The dynamic functional connectome: State-of-the-art and perspectives." Neuroimage 160 (2017): 41-54.
>
>
> > 6. More description should be added to the caption in Figure 1.
>
> We appreciate the reviewer for noticing the point that we have missed.
> A more thorough description of the Figure 1 will be provided in the revised manuscript as follows:
>
> Figure 1: Schematic illustration of the proposed method.
> (a) Overall framework of the STAGIN. A sequence of dynamic graph is first input to the GIN followed by GARO or SERO which produces a sequence of spatially attended graph representation vectors $\tilde{\mathbf{h} _ {G(t)}} $. Temporal attention is computed over $\tilde{\mathbf{h}} _ {G(t)}$ and the temporally attended graph representations are averaged to generate the final representation $\mathbf{h} _ {G _ {dyn}}$.
> (b) Attention-based READOUT modules. Both GARO and SERO compute spatial attention $\mathbf{z}_{\text{space}}$ with global average-pooled graph feature $\mathbf{h} _{G}$ as prior.
>
>
> > 7. Figure 2 does not seem to be a complete display. For example, there is no node feature $\mathbf{x}_{v}(t)$.
>
> We plan to update the Figure 2 to include $\mathbf{x}_{i}(t)$ which is a linearly mapped vector from the concatenation of $e_i$ and $\phi (t)$, as suggested.

---

> ### Author Response · Authors · 2021-08-10
> **Response to reviewer qvpg [1/2]**
>
> ## Main Review
>
> > 1. The innovation of the paper is limited. Both the attention mechanism and transformer are existing technologies. Authors do not highlight their proposed technology details and technical novelty. In addition, the paper lacks theoretical innovation.
>
> We agree with the authors that incorporating spatial / temporal attention to GNNs is a topic already being studied widely in the geometric deep learning field.
> Nevertheless, we would like to gently appeal that applying deep neural networks to the dynamic FC analysis is still not a well-established area which requires special considerations.
> For example, the dynamic brain graph has different characteristics from widely studied dynamic graphs (e.g. dynamic social network graphs).
> The dynamic brain graph does not have any change in existence (addition or deletion) of nodes through time.
> The timepoint is uniformly sampled depending on the repetition time (TR) during acquisition.
> There is no consensus yet on how the node features of the dynamic brain graph should be defined.
> Collecting one sample of fMRI data is very expensive, leading to lack of large-scale benchmark datasets.
> Data acquired from different institutions or different MRI devices is prone to creating non i.i.d samples, even with intensive preprocessing.
> These are some, but not all, of the hurdles to applying current SOTA methods off-the-shelf to the neuroimage analysis directly.
> We tried to stay tight on both ground of neuroimaging and deep learning, which might possibly seem to fall a little behind from either field.
> However, we believe that this work can definitely serve as an innovative research that provides guidance for researchers trying to bridge the gap between the two areas.
>
> Related studies on the attention mechanism, transformer, and the GNN will be provided in the revised manuscript to further highlight the differences and strengths of our proposed method as follows:
>
> - Attention in Graph Neural Networks
>
> Bringing attention to the GNNs is a topic that is being actively studied in the field of geometric deep learning [1].
> One of the most successful uses of attention is to compute the attention at edges of the graph and scale the importance of the links based on the attention when the features of the neighborhood node are aggregated [2,3].
> This attention-based aggregation scheme can often provide a performance gain in learning the representation of the input graphs.
> In our work, however, this scheme was not used in order to keep the original connectivity pattern on the later layers of the model.
>
>
> Another stream of applying attention to the GNNs comes with the motivation to define a pooling function on the graph domain.
> Unlike natural images which the data is in a regular grid, it is not straightforward to decide on what basis the coarsening should be carried out for graph structured data.
> The work by [4] addressed this problem by projecting the node feature vectors into a learnable parameter vector $\mathbf{p}$ and selecting the top k nodes based on the projected score.
> Subsequent studies have expanded this idea further to include local graph structures in the node scoring through variants of the graph convolution layers [5,6,7].
> These graph pooling methods have in common that they exploit learned relative scores across the vertices of the graph, which is closely related to the spatial attention modules that we propose in Section 3.2.
> Some works have already been aware that the appropriate use of node-wise attention can improve performance of downstream tasks [8,9].
> We address the issue that previous pooling methods tend to score attention based on randomly initialized parameters or local graph structures, which may be suboptimal for graph classification tasks that require taking the whole graph feature into account.
> To this end, we propose two attention-based READOUT modules that exploit global average-pooled whole graph feature as prior information to score attention across the nodes.
>
> - Graph Neural Network on Dynamic Graphs
>
> Many networks that arise around us are inherently dynamic, with the changes in the existence of nodes and edges over time.
> Learning the representation of dynamic graphs has piqued the interest of researchers and has led to development of methods that can embed dynamic graphs using their time information [10].
> Methods that incorporate attention for learning the representation of dynamic graphs have also been proposed [11,12].
> However, it is not easy to apply these techniques directly to the dynamic brain graphs because of the different inherent properties of the dynamic brain graphs that do not include any addition or deletion of nodes and that are sampled uniformly over time.
> Nonetheless, our work is inspired by these earlier studies, particularly for the encoding of temporal information and their concatenation to the node features, proposed in Section 3.1 [11,12].
>
> [1] Lee, John Boaz, et al. "Attention models in graphs: A survey." ACM Transactions on Knowledge Discovery from Data (TKDD) 13.6 (2019): 1-25.\
> [2] Veličković, Petar, et al. "Graph attention networks." arXiv preprint arXiv:1710.10903 (2017).\
> [3] Brody, Shaked, Uri Alon, and Eran Yahav. "How Attentive are Graph Attention Networks?." arXiv preprint arXiv:2105.14491 (2021).\
> [4] Gao, Hongyang, and Shuiwang Ji. "Graph u-nets." international conference on machine learning. PMLR, 2019.\
> [5] Lee, Junhyun, Inyeop Lee, and Jaewoo Kang. "Self-attention graph pooling." International Conference on Machine Learning. PMLR, 2019.\
> [6] Ranjan, Ekagra, Soumya Sanyal, and Partha Talukdar. "Asap: Adaptive structure aware pooling for learning hierarchical graph representations." Proceedings of the AAAI Conference on Artificial Intelligence. Vol. 34. No. 04. 2020.\
> [7] Knyazev, Boris, Graham W. Taylor, and Mohamed R. Amer. "Understanding attention and generalization in graph neural networks." arXiv preprint arXiv:1905.02850 (2019).\
> [8] Yan, Yichao, et al. "Learning multi-attention context graph for group-based re-identification." IEEE Transactions on Pattern Analysis and Machine Intelligence (2020).\
> [9] Fan, Xiaolong, et al. "Structured self-attention architecture for graph-level representation learning." Pattern Recognition 100 (2020): 107084.\
> [10] Nguyen, Giang Hoang, et al. "Continuous-time dynamic network embeddings." Companion Proceedings of the The Web Conference 2018. 2018.\
> [11] Xu, Da, et al. "Inductive representation learning on temporal graphs." arXiv preprint arXiv:2002.07962 (2020).\
> [12] Rossi, Emanuele, et al. "Temporal graph networks for deep learning on dynamic graphs." arXiv preprint arXiv:2006.10637 (2020).
>
>
>
>
> > 2. The authors claim that the proposed method achieves SOTA performance. The authors introduced a lot of related work in the Introduction, but only used 2 baseline methods. The authors should increase the baseline methods and report the method parameters and running time, etc.
>
> Thank you very much for the important comment.
> We are under implementing and experimenting with the baseline methods which could not have been fully completed within our response period due to resource constraints.
> We apologize for the inconvenience and will update the results including method parameters and running time as soon as they become available.
> For a preliminary comparison of the performance metrics before the final metrics are provided, we summarize the reported performance from the papers that used the HCP dataset.
>
> - Gender classification on HCP-Rest
>
> |      Method     |   FC    |   Accuracy   |   Reference |
> |---|---|---|---|
> |      *STAGIN     | dynamic | 88.20 (1.33) | Ours |
> |      ST-GCN     | dynamic | 83.7 ()      |  Gadgil (2020) |
> | TCN-GN-DiffPool | static  | 76 (1.0)     |  Azevedo (2020) |
> |       GIN       | static  | 84.61 (2.9)  |  Kim (2020) |
> |       GCN       | static  | 83.98 (3.2)  |  Arslan (2018) / Kim (2020) |
>
> We plan to implement other GNN models which have been applied to fMRI data but not for HCP-Rest gender classification.
>
> - Task decoding on HCP-Task
>
> |      Method     |         FC         |   Accuracy   | Reference |
> |---|---|---|---|
> |      *STAGIN     |       dynamic      | 99.19 (0.20) | Ours |
> |      BAnD++     | none (voxel-based) | 97.2 (0.57)  | Nguyen (2020) |
> |      BAnD       | none (voxel-based) | 97.0 (0.37)  | Nguyen (2020) |
> |      r-BAnD     |       dynamic      | 98.90 (0.27) | Ours for comparison |
> |      ST-GCN     |       dynamic      |              |  |
>
> We plan to implement ST-GCN on HCP-Task based on reviewers' suggestion
>
>
> > 3. The effect of hyperparameters on the results is not discussed. For example, $\lambda$ and edge threshold (30%).
>
> We find that the proposed method tend to be robust to hyperparameter changes, including $\lambda$ and edge threshold.
>
> |   $\lambda$    |   Accuracy   |      AUROC      |
> |---|---|---|
> | $1.0 \times 10^{-4}$ | 87.46 (2.56) | 0.9336 (0.0179) |
> | *$1.0 \times 10^{-5}$ | 88.20 (1.33) | 0.9296 (0.0187) |
> | $1.0 \times 10^{-6}$ | 88.10 (2.08) | 0.9347 (0.0194) |
>
>
> | Edge threshold |   Accuracy   |      AUROC      |
> |---|---|---|
> |   20%    | 88.01 (2.81) | 0.9304 (0.0220) |
> |   *30%   | 88.20 (1.33) | 0.9296 (0.0187) |
> |   40%    | **89.02** (1.80) | **0.9408** (0.0110) |
>
> It should be noted that the edge threshold 40% resulted in better accuracy and AUROC than the reported metric from our submission, which will be discussed in the updated manuscript.
> We appreciate the suggestion from the reviewer, which helped us improve our method further.

---

> > ### Comment · Reviewer_qvpg · 2021-08-26
> > **The innovation of the paper is limited**
> >
> > Thank you for your reply.
> >
> > As I mentioned before, the innovation of the paper is limited. Both the attention mechanism and transformer are existing technologies. Authors do not highlight their proposed technology details and technical novelty. In addition, the paper lacks theoretical innovation. The proposed model is just a simple combination of existing technologies.
> >
> > The authors only used 2 baseline methods. During the one-month response period, there were no updates. I have concerns about the performance of the method (such as running time). In addition, there is no detailed response to the parameters of the method.
> >
> > Simple visualization does not mean interpretability in the true sense. Some other inference methods can be applied, such as Bayesian neural networks.

---

> > > ### Author Response · Authors · 2021-08-27
> > > **Response to reviewer qvpg**
> > >
> > >
> > >
> > > > As I mentioned before, the innovation of the paper is limited. Both the attention mechanism and transformer are existing technologies. Authors do not highlight their proposed technology details and technical novelty. In addition, the paper lacks theoretical innovation. The proposed model is just a simple combination of existing technologies.
> > >
> > > We express our gratitude to the reviewer for the effort on reviewing our paper and respect the comments regarding our work.
> > >
> > > > The authors only used 2 baseline methods. During the one-month response period, there were no updates. I have concerns about the performance of the method (such as running time). In addition, there is no detailed response to the parameters of the method.
> > >
> > > The baseline methods will be finalized within next week, which we had difficulty in replicating the results of the work by [3] causing the delay. We are suspecting that this is related to a severe sensitivity in hyper-parameters of the model suggested by the Figure 3 of [3]. We again apologize for the inconvenience. The preliminary comparative result tables will be provided at the last part of this response.
> > >
> > >
> > > The number of parameters of our method is 1,209,804 in total, with 203,520 for the timestamp encoder, 134,148 for the GIN, 273,472 for the READOUT module SERO, 529,920 for the Transformer Encoder, and 68,744 for the linear mapping layers.
> > > It takes 1.68s for classifying one fMRI timeseries data, which the bottleneck comes from constructing the dynamic FC matrices, but not the model itself.
> > > We expect that this inference time can be much reduced with more efficient approach for dynamic brain graph construction.
> > >
> > > > Simple visualization does not mean interpretability in the true sense. Some other inference methods can be applied, such as Bayesian neural networks.
> > >
> > > We will try to incorporate Bayesian approach in our future works. We agree with the reviewer that Bayesian approach can help understand the data even better, which provides uncertainty of the model computation.
> > >
> > > ---
> > > ## Preliminary comparative study results
> > >
> > > - Gender classification on HCP-Rest
> > >
> > > |      Method     |   FC    |   Accuracy   | AUROC  | # Parameters |
> > > |---|---|---|---|---|
> > > |      *STAGIN-SERO-40     | dynamic | 89.02 (1.80) | 0.9408 (0.0110)  | 1,209,804 |
> > > |      *STAGIN-SERO     | dynamic | 88.20 (1.33) | 0.9296 (0.0187)  | 1,209,804 |
> > > |      *STAGIN-GARO     | dynamic | 87.01 (3.00) | 0.9151 (0.0258)   | 1,068,426 |
> > > |      ST-GCN     | dynamic |  76.95 (3.00)    |   0.8545 (0.0316)     | 355,042 |
> > > | TCN-GN-DiffPool | static  | 53.70 (2.40)      |  0.5580 (0.0246)       | 849,507 |
> > > |       GIN       | static  | 81.34 (2.40)  |  0.8955 (0.0237)   | 169,996 |
> > > |       GCN       | static  | 80.79 (2.00)  |  0.8741 (0.0174)  | 101,896 |
> > > |       GraphSAGE       | static  |  75.48 (1.97)   |  0.8237 (0.0228)    | 202,248 |
> > > |       ChebGCN       | static  |  77.76 (2.09)  |  0.8582 (0.0233)    | 704,008 |
> > >
> > > STAGIN-SERO-40 refers to STAGIN-SERO with dynamic FC graph sparsity set to 40\%
> > >
> > > - Task decoding on HCP-Task
> > >
> > > |      Method     |         FC         |   Accuracy   | # Parameters |
> > > |---|---|---|---|
> > > |      *STAGIN-SERO     |       dynamic      | 99.19 (0.20) | 1,209,804 |
> > > |      *STAGIN-GARO     |       dynamic      | 99.02 (0.17) |  1,068,426 |
> > > |      BAnD++     | none (voxel-based) | *97.2 (0.57)*  | |
> > > |      BAnD       | none (voxel-based) | *97.0 (0.37)*  | |
> > > |      r-BAnD     |       dynamic      | 98.90 (0.27) | 664,068
> > > |      ST-GCN     |       dynamic      | 98.92 (0.18) | 355,042
> > > |       GIN       | static  | 93.87 (0.66)   | 169,996 |
> > > |       GCN       | static  | 45.07 (1.63) | 101,896 |
> > > |       GraphSAGE       | static  |  54.52 (0.97)   | 202,248 |
> > > |       ChebGCN       | static  |  73.06 (0.68)   |  704,008 |
> > >
> > >
> > > It can be seen that static FC methods have difficulty in properly classifying the task types. The result is not very surprising since the sub-task timing information is completely lost in static FC methods, which can be a critical disadvantage in task classification.
> > >
> > > Italic numbers are numbers taken from the paper proposing the method, which is under re-implementation.

---

> ### Author Response · Authors · 2021-09-05
> **Response to reviewer qvpg - Comparative experiments**
>
> We provide our comparative experiment results as follows.
>
> - Gender classification on HCP-Rest
>
> |      Method     |   FC    |   Accuracy   | AUROC  | # Parameters |
> |---|---|---|---|---|
> |      *STAGIN-SERO-40     | dynamic | 89.02 (1.80) | 0.9408 (0.0110)  | 1,209,804 |
> |      *STAGIN-SERO     | dynamic | 88.20 (1.33) | 0.9296 (0.0187)  | 1,209,804 |
> |      *STAGIN-GARO     | dynamic | 87.01 (3.00) | 0.9151 (0.0258)   | 1,068,426 |
> |      ST-GCN     | dynamic |  76.95 (3.00)    |   0.8545 (0.0316)     | 355,042 |
> | TCN-GN-DiffPool | static  | 53.70 (2.40)      |  0.5580 (0.0246)       | 849,507 |
> |       GIN       | static  | 81.34 (2.40)  |  0.8955 (0.0237)   | 169,996 |
> |       GCN       | static  | 80.79 (2.00)  |  0.8741 (0.0174)  | 101,896 |
> |       GraphSAGE       | static  |  75.48 (1.97)   |  0.8237 (0.0228)    | 202,248 |
> |       ChebGCN       | static  |  77.76 (2.09)  |  0.8582 (0.0233)    | 704,008 |
>
> Asterisks indicate our proposed models.
> STAGIN-SERO-40 refers to STAGIN-SERO with dynamic FC graph sparsity set to 40\%
>
> - Task decoding on HCP-Task
>
> |      Method     |         FC         |   Accuracy   | # Parameters |
> |---|---|---|---|
> |      *STAGIN-SERO     |       dynamic      | 99.19 (0.20) | 1,209,804 |
> |      *STAGIN-GARO     |       dynamic      | 99.02 (0.17) |  1,068,426 |
> |      BAnD++     | none (voxel-based) | 97.29 (0.46)  | 2,010,176 |
> |      BAnD       | none (voxel-based) | 95.77 (0.65) | 2,010,176 |
> |      r-BAnD     |       dynamic      | 98.90 (0.27) | 664,068 |
> |      ST-GCN     |       dynamic      | 98.92 (0.18) | 355,042 |
> |       GIN       | static  | 93.87 (0.66)   | 169,996 |
> |       GCN       | static  | 45.07 (1.63) | 101,896 |
> |       GraphSAGE       | static  |  54.52 (0.97)   | 202,248 |
> |       ChebGCN       | static  |  73.06 (0.68)   |  704,008 |
>
> It can be seen that static FC methods have difficulty in properly classifying the task types. The result is not very surprising since the sub-task timing information is completely lost in static FC methods, which can be a critical disadvantage in task classification.

---

> ### Author Response · Authors · 2021-09-13
> **Response to reviewer qvpg - Comparative experiments [2]**
>
> A new study was recently published as a preprint which attempted to apply MS-G3D to the resting-state fMRI data [1].
> MS-G3D is an extension of the ST-GCN model, developed for action recognition [2].
> We have re-experimented the model with the code provided by the authors (https://github.com/metrics-lab/ST-fMRI/), but revised the training phase to not early stop based on the test set which can significantly exaggerate the test performance.
> The model required 134,039,479 parameters to process ROI-timeseries generated from the Schaefer400 atlas which was over 110$\times$ the size of our proposed STAGIN-SERO, resulting in OOM with our same experimental settings.
> Accuracy of the MS-G3D on HCP-Rest resulted in $79.16 \pm 2.53 \%$ with ICA-extracted ROI-timeseries with 22 nodes.
>
> |      Method     |   FC    |   Accuracy   | AUROC  | # Parameters |
> |---|---|---|---|---|
> |      *STAGIN-SERO-40     | dynamic | 89.02 (1.80) | 0.9408 (0.0110)  | 1,209,804 |
> |      MS-G3D     | dynamic | OOM |  OOM | 134,039,479 |
> |      MS-G3D (22 nodes)     | dynamic | 79.16 (2.53) | 0.8912 (0.0329)   | 3,045,283 |
>
> We cautiously assume that the unexpectedly high accuracy reported in the paper can possibly be related to peeking of the test dataset during the training phase.
>
> [1] Dahan, Simon, et al. "Improving Phenotype Prediction using Long-Range Spatio-Temporal Dynamics of Functional Connectivity." arXiv preprint arXiv:2109.03115 (2021). \
> [2] Liu, Ziyu, et al. "Disentangling and unifying graph convolutions for skeleton-based action recognition." Proceedings of the IEEE/CVF conference on computer vision and pattern recognition. 2020.

---

### Official Review · Reviewer_EUFS · 2021-07-16

**Rating:** 6
**Confidence:** 4

**Summary:**

The paper proposes an attention based GNN method, which it names STAGIN. The new method handles dynamic FC matrices for gender and task classification using fMRI data and learns to focus on important time points and brain regions based on attention. The method achieves better accuracy and AUC for classification compared to the relevant competing methods that STAGIN is compared against. The temporal attention matches with the task-relevant time points in the input data.


**Limitations And Societal Impact:**

Addressed

**Main Review:**

The main contribution of the paper is in handling dynamic FC matrices and finding important time points and brain regions relative to the downstream task of classification. The paper uses GNN to model brain data into nodes representing brain regions and edges representing functional connectivity. The paper shows higher accuracy and AUC scores for the task of gender and task classification compared to the methods mentioned in the paper. The paper has the following major and minor drawbacks.

## Major Drawbacks:

1.  The idea of applying attention to graph networks is not novel as there have been many published studies for this e.g. GAT, GATV2 etc. The paper applies attention to the published method GIN but does not compare similarities or differences with existing such approaches e.g. “Understanding Attention and Generalization in Graph Neural Networks” ChebyGIN.
2.  The author(s) propose a novel attention based Readout method and add it in GIN to create a single embedding for the complete graph. The authors propose two variations of the method. Again, such methods are already well established e.g. topKpooling, SAG pooling, ASA pooling. The author(s) do not discuss or compare against such methods and rather mention the original method of GIN. The author(s) uses temporal data with a sliding window approach to create dynamic FC matrices but do not compare against methods which handle temporal data and dynamic graphs e.g. TGAT, TGN-attention, CTDNE etc. These methods may not have been applied for the same downstream classification task but still a comparison with them would reveal if the proposed method is improving over existing analogous approaches. The paper is well written and experiments are thorough and well structured, but it is unclear whether a technical novelty is introduced or largely existing methods are applyed to fMRI data for gender/task classification. (Also se item 2 in the minor drawbacks)
3.  The paper needs to report results from much more published studies, especially those which handle dynamic/temporal data as mentioned above. The author(s) also need to show how their attention mechanism is better than existing methods of attention in GNNs.
4.  The author(s) need to mention how they got the result of BAND and BaND++ and whether the presented comparison is made on the same data with the same processing. BAND and BaND++ are applied to 4D data of voxels and not on region based data and thus would be at a statistical disadvantage if compared with the method that uses region based aggregation.

## Minor Drawbacks:

1.  In Fig 3, the left three clusters namely "6,2,7" have more females than male but DMN does not have hyper connectivity compared to SMN. Whereas the last three clusters with more male do show hyperconnectivity in SMN. This should be explained in more detail.
2.  Since the FC is not learned rather is computed using covariance, the author(s) need to show how the connectivity of DMN and SMN looks in time points which are not attended by the model. It is quite possible that the DMN and SMN connectivity is different for males and females per the referenced literature in all time points and not just the attended ones. If this is true, then the value provided by the presented model is even less clear.
3.  The author(s) mention at line 237 that it's average of columns but looks like average of rows. Also the index j is not specified.
4.  The author(s) should explain why the 30th percentile was chosen when creating the adjacency matrix using FC.
5.  The author(s) should be more clear with what they mean by dynamics in this work. Notably, 1) The regions/nodes do not change through time 2) The number of edges also remain fixed. Please be specific and explicitly explain what does change?
6.  The ablation study in the appendix shows that attention doesn't improve AUC results by much. Also, the last row for the main atlas shows very high accuracy and AUC without any of the four components. This needs more explanation as it looks like the underlying classification task is not difficult.

**Time Spent Reviewing:**

7

---

> ### Author Response · Authors · 2021-08-10
> **Response to reviewer EUFS [2/2]**
>
> > 4. The author(s) need to mention how they got the result of BAND and BaND++ and whether the presented comparison is made on the same data with the same processing.
>
> The accuracy of BAnD and BAnD++ were taken from the literature, which we plan to re-implement with our dataset upon revision.
>
> > BAND and BaND++ are applied to 4D data of voxels and not on region based data and thus would be at a statistical disadvantage if compared with the method that uses region based aggregation.
>
> In order to make a fair comparison regarding the statistical disadvantage of voxel-based feature extraction, we needed a different feature extracting module than the 3D-ResNet.
> We implemented a new "region-based BAnD (r-BAnD)" by using GIN (without attention-based READOUT) instead of the 3D-ResNet for the region-based comparison.
> Accuracy of r-BAnD resulted in an accuracy of $98.90 \pm 0.27$%, suggesting that our method shows superior performance even when the statistical disadvantages are matched.
>
>
>
> ### Minor Drawbacks
> > 1. In Fig 3, the left three clusters namely "6,2,7" have more females than male but DMN does not have hyper connectivity compared to SMN. Whereas the last three clusters with more male do show hyperconnectivity in SMN. This should be explained in more detail.
>
> Neuroimaging evidences suggest hyperconnectivity of DMN and hypoconnectivity of SMN in female subjects compared to male subjects, but not between DMN and SMN.
> Therefore, we believe that the SMN having stronger connection than the DMN does not stand against our interpretation.
> We will describe the interpretation more clearly in the revised manuscript.
>
> > 2. Since the FC is not learned rather is computed using covariance, the author(s) need to show how the connectivity of DMN and SMN looks in time points which are not attended by the model. It is quite possible that the DMN and SMN connectivity is different for males and females per the referenced literature in all time points and not just the attended ones. If this is true, then the value provided by the presented model is even less clear.
>
> We plotted average DMN and SMN connectivity matrix uniformly subsampled over time / subject which is irrespective of any attention effect.
> Average connectivity pattern between female and male subjects was almost identical in both the DMN and the SMN, with minimal hyperconnectivity of DMN and hypoconnectivity of SMN present in the female subjects compared to male subjects.
> This average pattern again confirms the validity of our method by showing that our method can capture the small difference between the two groups that is present in the dynamic FC graph, and exploit the captured information for classification.
>
>
> > 3. The author(s) mention at line 237 that it's average of columns but looks like average of rows. Also the index j is not specified.
>
> We apologize for the misleading typo. We will correct the notation as suggested.
>
>
> > 4. The author(s) should explain why the 30th percentile was chosen when creating the adjacency matrix using FC.
>
> We followed a related work by Kim and Ye (2020) to set the sparsity to 30% in the original paper.
> We experimented with other edge threshold values and find that the proposed method tend to be robust to the hyperparameter change:
>
> | Edge threshold |   Accuracy   |      AUROC      |
> |---|---|---|
> |   20%    | 88.01 (2.81) | 0.9304 (0.0220) |
> |   *30%   | 88.20 (1.33) | 0.9296 (0.0187) |
> |   40%    | **89.02** (1.80) | **0.9408** (0.0110) |
>
> It should be noted that the edge threshold 40% resulted in better accuracy and AUROC than the reported metric from our submission, which will be discussed in the updated manuscript.
> We appreciate the suggestion from the reviewer, which helped us improve our method further.
>
>
>
> > 5. The author(s) should be more clear with what they mean by dynamics in this work. Notably, 1) The regions/nodes do not change through time 2) The number of edges also remain fixed. Please be specific and explicitly explain what does change?
>
>
> The dynamic functional connectivity indeed refers to the change in the edge strength of the brain connectome through time, which does not accompany any change in the existence of vertices (e.g. addition, deletion).
> However, the functional connectivity between every regions of the brain changes over time and this change in connectivity is believed to carry important signal of the brain [13,14].
> In this work, the adjacency matrix is binarized so the connection between the regions of the brain fluctuate between connected and unconnected state over time while the number of nodes and connections are fixed.
> Figure 2 (b) partly demonstrates this characteristic of dynamic FC by showing that the number of edges are fixed to 4 at any timepoint $t$, but the connectivity pattern changes from $\{(1,2), (2,3), (2,5), (3,4)\}$ to $\{(1,4), (1,5), (2,3), (2,5)\}$.
>
> [13] Hutchison, R. Matthew, et al. "Dynamic functional connectivity: promise, issues, and interpretations." Neuroimage 80 (2013): 360-378.\
> [14] Preti, Maria Giulia, Thomas AW Bolton, and Dimitri Van De Ville. "The dynamic functional connectome: State-of-the-art and perspectives." Neuroimage 160 (2017): 41-54.
>
>
> > 6. The ablation study in the appendix shows that attention doesn't improve AUC results by much. Also, the last row for the main atlas shows very high accuracy and AUC without any of the four components. This needs more explanation as it looks like the underlying classification task is not difficult.
>
> We agree with the reviewer that the spatio-temporal attention does not improve AUROC as significantly as the timestamp encoding.
> We will discuss in the revised manuscript that the performance gain is mainly related to the timestamp encoding, while the spatio-temporal attention can provide minor improvements.
> While improvement in AUROC may not be as significant, we still claim that attention is important as they provide spatio-temporal explainability of the model, which is very crucial for functional neuroimaging methods.

---

> > ### Comment · Area_Chair_4Dqm · 2021-08-25
> > **Udpdate your evaluation ?**
> >
> > Dear Reveiwer EUFS,
> >
> > Do you wish to comment about the detailed author's response, or do you want to update your evaluation ?
> > Did it address your concerns properly or does there remain some issues ?
> > Thx for your contribution,

---

> > > ### Comment · Reviewer_EUFS · 2021-08-26
> > > **Looking forward to the updated paper**
> > >
> > > The authors will indeed have an improved paper after updating it with the experiments that they have described. The accuracy improvements are marginal, but the approach is potentially useful and interesting despite some reservations about its novelty to the NeurIPS community. I am increasing my score and wishing the authors good luck!

---

> > > > ### Author Response · Authors · 2021-08-27
> > > > **Response to reviewer EUFS**
> > > >
> > > > We express our sincere gratitude to the reviewer for providing detailed comments which helped us greatly improve our work.
> > > > The experiments will be finalized within next week, and hope to share our improved work in the conference.

---

> ### Author Response · Authors · 2021-08-10
> **Response to reviewer EUFS [1/2]**
>
> ## Main Review
> ### Major Drawbacks
>
> >1. The idea of applying attention to graph networks is not novel as there have been many published studies for this e.g. GAT, GATV2 etc. The paper applies attention to the published method GIN but does not compare similarities or differences with existing such approaches e.g. “Understanding Attention and Generalization in Graph Neural Networks” ChebyGIN.
> >2. The author(s) propose a novel attention based Readout method and add it in GIN to create a single embedding for the complete graph. The authors propose two variations of the method. Again, such methods are already well established e.g. topKpooling, SAG pooling, ASA pooling. The author(s) do not discuss or compare against such methods and rather mention the original method of GIN. The author(s) uses temporal data with a sliding window approach to create dynamic FC matrices but do not compare against methods which handle temporal data and dynamic graphs e.g. TGAT, TGN-attention, CTDNE etc. These methods may not have been applied for the same downstream classification task but still a comparison with them would reveal if the proposed method is improving over existing analogous approaches. The paper is well written and experiments are thorough and well structured, but it is unclear whether a technical novelty is introduced or largely existing methods are applyed to fMRI data for gender/task classification. (Also se item 2 in the minor drawbacks)
>
> Thank you very much for the important comments.
> The mentioned prior works are indeed closely related to our work, and we plan to discuss them in the revised manuscript as follows:
>
>
> - Attention in Graph Neural Networks
>
> Bringing attention to the GNNs is a topic that is being actively studied in the field of geometric deep learning [1].
> One of the most successful uses of attention is to compute the attention at edges of the graph and scale the importance of the links based on the attention when the features of the neighborhood node are aggregated [2,3].
> This attention-based aggregation scheme can often provide a performance gain in learning the representation of the input graphs.
> In our work, however, this scheme was not used in order to keep the original connectivity pattern on the later layers of the model.
>
>
> Another stream of applying attention to the GNNs comes with the motivation to define a pooling function on the graph domain.
> Unlike natural images which the data is in a regular grid, it is not straightforward to decide on what basis the coarsening should be carried out for graph structured data.
> The work by [4] addressed this problem by projecting the node feature vectors into a learnable parameter vector $\mathbf{p}$ and selecting the top k nodes based on the projected score.
> Subsequent studies have expanded this idea further to include local graph structures in the node scoring through variants of the graph convolution layers [5,6,7].
> These graph pooling methods have in common that they exploit learned relative scores across the vertices of the graph, which is closely related to the spatial attention modules that we propose in Section 3.2.
> Some works have already been aware that the appropriate use of node-wise attention can improve performance of downstream tasks [8,9].
> We address the issue that previous pooling methods tend to score attention based on randomly initialized parameters or local graph structures, which may be suboptimal for graph classification tasks that require taking the whole graph feature into account.
> To this end, we propose two attention-based READOUT modules that exploit global average-pooled whole graph feature as prior information to score attention across the nodes.
>
> - Graph Neural Network on Dynamic Graphs
>
> Many networks that arise around us are inherently dynamic, with the changes in the existence of nodes and edges over time.
> Learning the representation of dynamic graphs has piqued the interest of researchers and has led to development of methods that can embed dynamic graphs using their time information [10].
> Methods that incorporate attention for learning the representation of dynamic graphs have also been proposed [11,12].
> However, it is not easy to apply these techniques directly to the dynamic brain graphs because of the different inherent properties of the dynamic brain graphs that do not include any addition or deletion of nodes and that are sampled uniformly over time.
> Nonetheless, our work is inspired by these earlier studies, particularly for the encoding of temporal information and their concatenation to the node features, proposed in Section 3.1 [11,12].
>
> [1] Lee, John Boaz, et al. "Attention models in graphs: A survey." ACM Transactions on Knowledge Discovery from Data (TKDD) 13.6 (2019): 1-25.\
> [2] Veličković, Petar, et al. "Graph attention networks." arXiv preprint arXiv:1710.10903 (2017).\
> [3] Brody, Shaked, Uri Alon, and Eran Yahav. "How Attentive are Graph Attention Networks?." arXiv preprint arXiv:2105.14491 (2021).\
> [4] Gao, Hongyang, and Shuiwang Ji. "Graph u-nets." international conference on machine learning. PMLR, 2019.\
> [5] Lee, Junhyun, Inyeop Lee, and Jaewoo Kang. "Self-attention graph pooling." International Conference on Machine Learning. PMLR, 2019.\
> [6] Ranjan, Ekagra, Soumya Sanyal, and Partha Talukdar. "Asap: Adaptive structure aware pooling for learning hierarchical graph representations." Proceedings of the AAAI Conference on Artificial Intelligence. Vol. 34. No. 04. 2020.\
> [7] Knyazev, Boris, Graham W. Taylor, and Mohamed R. Amer. "Understanding attention and generalization in graph neural networks." arXiv preprint arXiv:1905.02850 (2019).\
> [8] Yan, Yichao, et al. "Learning multi-attention context graph for group-based re-identification." IEEE Transactions on Pattern Analysis and Machine Intelligence (2020).\
> [9] Fan, Xiaolong, et al. "Structured self-attention architecture for graph-level representation learning." Pattern Recognition 100 (2020): 107084.\
> [10] Nguyen, Giang Hoang, et al. "Continuous-time dynamic network embeddings." Companion Proceedings of the The Web Conference 2018. 2018.\
> [11] Xu, Da, et al. "Inductive representation learning on temporal graphs." arXiv preprint arXiv:2002.07962 (2020).\
> [12] Rossi, Emanuele, et al. "Temporal graph networks for deep learning on dynamic graphs." arXiv preprint arXiv:2006.10637 (2020).
>
>
>
> >3. The paper needs to report results from much more published studies, especially those which handle dynamic/temporal data as mentioned above.
>
> We are under implementing and experimenting with the baseline methods which could not have been fully completed within our response period due to resource constraints.
> We apologize for the inconvenience and will update the results as soon as they become available.
> For a preliminary comparison of the performance metrics before the final metrics are provided, we summarize the reported performance from the papers that used the HCP dataset.
>
> - Gender classification on HCP-Rest
>
> |      Method     |   FC    |   Accuracy   |   Reference |
> |---|---|---|---|
> |      *STAGIN     | dynamic | 88.20 (1.33) | Ours |
> |      ST-GCN     | dynamic | 83.7 ()      |  Gadgil (2020) |
> | TCN-GN-DiffPool | static  | 76 (1.0)     |  Azevedo (2020) |
> |       GIN       | static  | 84.61 (2.9)  |  Kim (2020) |
> |       GCN       | static  | 83.98 (3.2)  |  Arslan (2018) / Kim (2020) |
>
> We plan to implement other GNN models which have been applied to fMRI data but not for HCP-Rest gender classification.
>
> - Task decoding on HCP-Task
>
> |      Method     |         FC         |   Accuracy   | Reference |
> |---|---|---|---|
> |      *STAGIN     |       dynamic      | 99.19 (0.20) | Ours |
> |      BAnD++     | none (voxel-based) | 97.2 (0.57)  | Nguyen (2020) |
> |      BAnD       | none (voxel-based) | 97.0 (0.37)  | Nguyen (2020) |
> |      r-BAnD     |       dynamic      | 98.90 (0.27) | Ours for comparison |
> |      ST-GCN     |       dynamic      |              |  |
>
> We plan to implement ST-GCN on HCP-Task based on reviewers' suggestion
>
>
> > The author(s) also need to show how their attention mechanism is better than existing methods of attention in GNNs.
>
> While the motivation may have been different, we agree with the reviewer that our attention-based READOUT functions share methodological similarity with graph pooling methods, which score and rank each nodes within the graph for the selection of important nodes.
> We experimented on replacing our attention-based READOUT functions with some well-known graph pooling methods ([TopKPooling](https://pytorch-geometric.readthedocs.io/en/latest/modules/nn.html#torch_geometric.nn.pool.TopKPooling), [SAGPooling](https://pytorch-geometric.readthedocs.io/en/latest/modules/nn.html#torch_geometric.nn.pool.SAGPooling), [ASAPooling](https://pytorch-geometric.readthedocs.io/en/latest/modules/nn.html#torch_geometric.nn.pool.ASAPooling) from the [PyTorch Geometric](https://pytorch-geometric.readthedocs.io/en/latest/index.html) package) without dropping any vertices for scoring the level of attention across the nodes.
> The results suggest that our attention-based READOUT functions perform better and more stable, with lower computational overload for our graph classification task.
>
> | READOUT |   Accuracy   |      AUROC      |
> |---|---|---|
> |   SERO  | 88.01 (2.81) | 0.9304 (0.0220) |
> |   TopKPooling  | 77.02 (10.94) | 0.8203 (0.1123) |
> |   SAGPooling   | OOM | OOM |
> |   ASAPooling   | OOM | OOM |
>
> We believe that the strength of our attention-based READOUT comes from taking the globally pooled graph feature ($\mathbf{H} \Phi_{\text{mean}}$) as a prior, which may represent the whole graph property better than a randomly initialized learnable vector (TopKPooling) or GNN aggregated close neighborhood information (SAGPooling, ASAPooling).
> Extending our attention-based READOUT functions for graph pooling can be an interesting future study, which was beyond the scope of this research.

---

> ### Author Response · Authors · 2021-09-05
> **Response to reviewer EUFS - Comparative experiments**
>
> We provide our comparative experiment results as follows.
>
> - Gender classification on HCP-Rest
>
> |      Method     |   FC    |   Accuracy   | AUROC  | # Parameters |
> |---|---|---|---|---|
> |      *STAGIN-SERO-40     | dynamic | 89.02 (1.80) | 0.9408 (0.0110)  | 1,209,804 |
> |      *STAGIN-SERO     | dynamic | 88.20 (1.33) | 0.9296 (0.0187)  | 1,209,804 |
> |      *STAGIN-GARO     | dynamic | 87.01 (3.00) | 0.9151 (0.0258)   | 1,068,426 |
> |      ST-GCN     | dynamic |  76.95 (3.00)    |   0.8545 (0.0316)     | 355,042 |
> | TCN-GN-DiffPool | static  | 53.70 (2.40)      |  0.5580 (0.0246)       | 849,507 |
> |       GIN       | static  | 81.34 (2.40)  |  0.8955 (0.0237)   | 169,996 |
> |       GCN       | static  | 80.79 (2.00)  |  0.8741 (0.0174)  | 101,896 |
> |       GraphSAGE       | static  |  75.48 (1.97)   |  0.8237 (0.0228)    | 202,248 |
> |       ChebGCN       | static  |  77.76 (2.09)  |  0.8582 (0.0233)    | 704,008 |
>
> Asterisks indicate our proposed models.
> STAGIN-SERO-40 refers to STAGIN-SERO with dynamic FC graph sparsity set to 40\%
>
> - Task decoding on HCP-Task
>
> |      Method     |         FC         |   Accuracy   | # Parameters |
> |---|---|---|---|
> |      *STAGIN-SERO     |       dynamic      | 99.19 (0.20) | 1,209,804 |
> |      *STAGIN-GARO     |       dynamic      | 99.02 (0.17) |  1,068,426 |
> |      BAnD++     | none (voxel-based) | 97.29 (0.46)  | 2,010,176 |
> |      BAnD       | none (voxel-based) | 95.77 (0.65) | 2,010,176 |
> |      r-BAnD     |       dynamic      | 98.90 (0.27) | 664,068 |
> |      ST-GCN     |       dynamic      | 98.92 (0.18) | 355,042 |
> |       GIN       | static  | 93.87 (0.66)   | 169,996 |
> |       GCN       | static  | 45.07 (1.63) | 101,896 |
> |       GraphSAGE       | static  |  54.52 (0.97)   | 202,248 |
> |       ChebGCN       | static  |  73.06 (0.68)   |  704,008 |
>
> It can be seen that static FC methods have difficulty in properly classifying the task types. The result is not very surprising since the sub-task timing information is completely lost in static FC methods, which can be a critical disadvantage in task classification.

---

> ### Author Response · Authors · 2021-09-13
> **Response to reviewer EUFS - Comparative experiments [2]**
>
> A new study was recently published as a preprint which attempted to apply MS-G3D to the resting-state fMRI data [1].
> MS-G3D is an extension of the ST-GCN model, developed for action recognition [2].
> We have re-experimented the model with the code provided by the authors (https://github.com/metrics-lab/ST-fMRI/), but revised the training phase to not early stop based on the test set which can significantly exaggerate the test performance.
> The model required 134,039,479 parameters to process ROI-timeseries generated from the Schaefer400 atlas which was over 110$\times$ the size of our proposed STAGIN-SERO, resulting in OOM with our same experimental settings.
> Accuracy of the MS-G3D on HCP-Rest resulted in $79.16 \pm 2.53 \%$ with ICA-extracted ROI-timeseries with 22 nodes.
>
> |      Method     |   FC    |   Accuracy   | AUROC  | # Parameters |
> |---|---|---|---|---|
> |      *STAGIN-SERO-40     | dynamic | 89.02 (1.80) | 0.9408 (0.0110)  | 1,209,804 |
> |      MS-G3D     | dynamic | OOM |  OOM | 134,039,479 |
> |      MS-G3D (22 nodes)     | dynamic | 79.16 (2.53) | 0.8912 (0.0329)   | 3,045,283 |
>
> We cautiously assume that the unexpectedly high accuracy reported in the paper can possibly be related to peeking of the test dataset during the training phase.
>
> [1] Dahan, Simon, et al. "Improving Phenotype Prediction using Long-Range Spatio-Temporal Dynamics of Functional Connectivity." arXiv preprint arXiv:2109.03115 (2021). \
> [2] Liu, Ziyu, et al. "Disentangling and unifying graph convolutions for skeleton-based action recognition." Proceedings of the IEEE/CVF conference on computer vision and pattern recognition. 2020.

---

### Official Review · Reviewer_RkTU · 2021-07-16

**Rating:** 7
**Confidence:** 5

**Summary:**

This paper proposes a GNN model (STAGIN) for learning from dynamic, spatio-temporal fMRI data. The model, based on GIN, takes a temporal sequence of brain graphs as input and includes a time encoding in the node features. The standard readout function is modified to incorporate spatial attention in two possible ways (scaled dot product attention and squeeze-excitation model). The series of graph features is then passed to a transformer-style self-attention encoder to incorporate temporal attention. Orthogonality of node feature vectors is also encouraged in the loss function. The method was tested on the public HPC datasets (~1100 subjects) to classify gender from resting state fMRI and decode task from task-based fMRI, and analysis of the temporal and spatial attention results was performed.

**Limitations And Societal Impact:**

Some limitations are addressed (threshold for determining attended regions), and they mention that the understanding brain connectome has both positive and negative societal impact, as linked to search for new biomarkers, e.g., for disease. However, it is not really clear what negative impact they mean by this, and I think some elaboration is needed.

**Main Review:**

Originality:
The authors present some new methods for learning dynamic graph representations in the context of brain connectomics. While prior work on methods for learning from temporally changing brain graphs are presented, I believe there is some discussion missing of prior work on the proposed methods of including attention in GNN networks, specifically for including attention in the readout layer (eg., [1,2]). Please discuss how the proposed methods differs from such prior work.

[1] Fan et al., Structured self-attention architecture for graph-level representation learning, 2020.
[2] Yan et al., Learning Multi-Attention Context Graph for Group-Based Re-Identification, 2020.

Quality:
This is a nice paper that brings in some often used modules (scaled dot attention / squeeze excitation, self attention) to the GNN model to analyze dynamic graph data, specifically 4D fMRI.  The methods seem appropriate and is a well thought-out piece of work, with some nice posthoc analysis of the attention results with respect to connectome understanding (e.g., visualizations showing the changes in spatial attention following the changes in task, visualizations of network differences for males/females). My primary critiques lie in the experimental methods. Some detailed comments:

1. In Sec. 3.2.3, the authors discuss the use of orthogonal regularization for the node features, and say they "find it desirable to encourage the orthogonality of H". I am not sure I understand the motivation behind the orthogonal constraint, compared to some other form of regularization - it would be nice for authors to elaborate on that.
2. My biggest concern lies in the experimental portion of the paper. The authors do include results of ablation experiments in the supplementary materials, but in the proposed method is compared to only 1 other approach for the gender classification task and 2 variations of 1 approach for the decoding task.  Furthermore, while it seems the result for the gender task was produced by the authors, the decoding results were taken from the literature, and it is unlikely that the exact same subjects and preprocessing step/experimental methods were used. Thus, I would say the numbers are informative but not directly comparable. Also, given that the ST-GCN result was produced by the authors, I don't know why this method wasn't also tested on the decoding task.
3. Authors appropriately list all the hyperparameter settings, and it seems that the same settings were used for all folds (no tuning). However, I am wondering about the setting for the time window length $\Gamma$. While authors give references to support their choice, in general window length in standard dynamic connectivity analysis is highly debated (actually as mentioned in one of their references) and the choice is generally arbitrary, with analysis results potentially greatly depending on the windowing. While the authors do provide a fairly complete study overall, I think it would be great to see the effect of different window sizes on the proposed method's results.
4. I am wondering for the k-means clustering of attended graphs, how/why the authors chose 7 clusters?

Clarity:
This is a well-written paper - it is well-organized with appropriate amount of background and the writing is very clear. Just a couple notes:
1. p.2, line 82: Authors wrote $f = g \circ q$ where $g$ is the GNN and $q$ is the transformer encoder - should be $q \circ g$
2. Fig. 3 caption: "attened" --> "attended"

Significance:
Improved dynamic analysis of fMRI data could be very significant, allowing for better understand of underlying neurological processes and disease, and I think this work would definitely be of interest to readers in this field. I think the proposed method for including spatial and temporal attention to GNN analysis could lead to others building off of such ideas.


**Time Spent Reviewing:**

4

---

> ### Author Response · Authors · 2021-08-10
> **Response to reviewer RkTU [2/2]**
>
>
> > 4. I am wondering for the k-means clustering of attended graphs, how/why the authors chose 7 clusters?
>
> The number of clusters k=7 was chosen following a well-known dynamic resting-state fMRI study by [13].
> We further experimented k=3 and k=5, and confirmed the same trend of DMN / SMN functional connectivity strength between the two genders.
> The k-means clustering result with k=3 and k=5 will be added to the revised manuscript.
>
> [13] Allen, Elena A., et al. "Tracking whole-brain connectivity dynamics in the resting state." Cerebral cortex 24.3 (2014): 663-676.
>
> ### Clarity
>
> >1. p.2, line 82: Authors wrote $f = g \circ q$ where $g$ is the GNN and $q$ is the transformer encoder - should be $q \circ g$
> >2. Fig. 3 caption: "attened" --> "attended"
>
> Thank you very much for the thorough and careful reading of our paper, and we apologize for the misleading typo.
> These will be fixed on the updated paper based on the reviewer's suggestion.
>
>
> ## Limitations And Societal Impact
> > 1. Some limitations are addressed (threshold for determining attended regions), and they mention that the understanding brain connectome has both positive and negative societal impact, as linked to search for new biomarkers, e.g., for disease. However, it is not really clear what negative impact they mean by this, and I think some elaboration is needed.
>
>
> We believe that all attempts to decode trait and state from human brain signal measurements have potential negative societal impact of abuse or misuse.
> For example, a method that can accurately decode what someone is thinking from fMRI signals can raise privacy concerns.
> Although our method is yet far behind the decoding capability that can be abused or misused, our research cannot still be free from these ethical considerations.
> This negative societal impact will be further elaborated in the revised manuscript.

---

> > ### Comment · Reviewer_RkTU · 2021-08-30
> > **Thanks for the authors' response**
> >
> > I thank the authors for their response, and appreciate the additional hyperparameter experiments and intention to add comparisons. Based on their response and other reviews, I keep my original rating (7). I agree with some points by other reviewers that the components are not necessarily novel in themselves, but the specific application and presented study with additional experimental comparisons I think would be of interest to the community.

---

> > > ### Author Response · Authors · 2021-08-31
> > > **Response to reviewer RkTU**
> > >
> > > We deeply appreciate the reviewer for providing detailed comments which helped us greatly improve this study. We also believe that the topic of learning representation of the brain connectome with deep neural networks has great potential impact, but there are still a number of practical barriers to bridging the gap between neuroimaging and deep learning. We look forward to sharing our work with the community to discuss and develop further ideas on the topic.

---

> ### Author Response · Authors · 2021-08-10
> **Response to reviewer RkTU [1/2]**
>
> ## Main Review
> ### Originality
> > 1. The authors present some new methods for learning dynamic graph representations in the context of brain connectomics. While prior work on methods for learning from temporally changing brain graphs are presented, I believe there is some discussion missing of prior work on the proposed methods of including attention in GNN networks, specifically for including attention in the readout layer (eg., [1,2]). Please discuss how the proposed methods differs from such prior work.
>
> The recommended prior works are indeed related to our work, and we thank the reviewer for the comment.
> Since other reviewers have also suggested adding related work on graph pooling and attention, we plan to discuss them together in the revised manuscript as follows:
>
> - Attention in Graph Neural Networks
>
> Bringing attention to the GNNs is a topic that is being actively studied in the field of geometric deep learning [1].
> One of the most successful uses of attention is to compute the attention at edges of the graph and scale the importance of the links based on the attention when the features of the neighborhood node are aggregated [2,3].
> This attention-based aggregation scheme can often provide a performance gain in learning the representation of the input graphs.
> In our work, however, this scheme was not used in order to keep the original connectivity pattern on the later layers of the model.
>
>
> Another stream of applying attention to the GNNs comes with the motivation to define a pooling function on the graph domain.
> Unlike natural images which the data is on a regular grid, it is not straightforward to decide on what basis the coarsening should be carried out for graph structured data.
> The work by [4] addressed this problem by projecting the node feature vectors into a learnable parameter vector $\mathbf{p}$ and selecting the top k nodes based on the projected score.
> Subsequent studies have expanded this idea further to include local graph structures in the node scoring through variants of the graph convolution layers [5,6,7].
> These graph pooling methods have in common that they exploit learned relative scores across the vertices of the graph, which is closely related to the spatial attention modules that we propose in Section 3.2.
> Some works have already been aware that the appropriate use of node-wise attention can improve performance of downstream tasks [8,9].
> We note that the previous pooling methods tend to score attention based on randomly initialized parameters or local graph structures, and  may be suboptimal for graph classification tasks that require taking the whole graph feature into account.
> Accordingly, we propose two attention-based READOUT modules that exploit global average-pooled whole graph feature as prior information to score attention across the nodes.
>
> - Graph Neural Network on Dynamic Graphs
>
> Many networks that arise around us are inherently dynamic, with the changes in the existence of nodes and edges over time.
> Learning the representation of dynamic graphs has piqued the interest of researchers and has led to development of methods that can embed dynamic graphs using their time information [10].
> Methods that incorporate attention for learning the representation of dynamic graphs have also been proposed [11,12].
> However, it is not easy to apply these techniques directly to the dynamic brain graphs because of the different inherent properties of the dynamic brain graphs that do not include any addition or deletion of nodes and that are sampled uniformly over time.
> Nonetheless, our work is inspired by these earlier studies, particularly for the encoding of temporal information and their concatenation to the node features, as proposed in Section 3.1 [11,12].
>
> [1] Lee, John Boaz, et al. "Attention models in graphs: A survey." ACM Transactions on Knowledge Discovery from Data (TKDD) 13.6 (2019): 1-25.\
> [2] Veličković, Petar, et al. "Graph attention networks." arXiv preprint arXiv:1710.10903 (2017).\
> [3] Brody, Shaked, Uri Alon, and Eran Yahav. "How Attentive are Graph Attention Networks?." arXiv preprint arXiv:2105.14491 (2021).\
> [4] Gao, Hongyang, and Shuiwang Ji. "Graph u-nets." international conference on machine learning. PMLR, 2019.\
> [5] Lee, Junhyun, Inyeop Lee, and Jaewoo Kang. "Self-attention graph pooling." International Conference on Machine Learning. PMLR, 2019.\
> [6] Ranjan, Ekagra, Soumya Sanyal, and Partha Talukdar. "Asap: Adaptive structure aware pooling for learning hierarchical graph representations." Proceedings of the AAAI Conference on Artificial Intelligence. Vol. 34. No. 04. 2020.\
> [7] Knyazev, Boris, Graham W. Taylor, and Mohamed R. Amer. "Understanding attention and generalization in graph neural networks." arXiv preprint arXiv:1905.02850 (2019).\
> [8] Yan, Yichao, et al. "Learning multi-attention context graph for group-based re-identification." IEEE Transactions on Pattern Analysis and Machine Intelligence (2020).\
> [9] Fan, Xiaolong, et al. "Structured self-attention architecture for graph-level representation learning." Pattern Recognition 100 (2020): 107084.\
> [10] Nguyen, Giang Hoang, et al. "Continuous-time dynamic network embeddings." Companion Proceedings of the The Web Conference 2018. 2018.\
> [11] Xu, Da, et al. "Inductive representation learning on temporal graphs." arXiv preprint arXiv:2002.07962 (2020).\
> [12] Rossi, Emanuele, et al. "Temporal graph networks for deep learning on dynamic graphs." arXiv preprint arXiv:2006.10637 (2020).
>
>
> ### Quality
> >1. In Sec. 3.2.3, the authors discuss the use of orthogonal regularization for the node features, and say they "find it desirable to encourage the orthogonality of H". I am not sure I understand the motivation behind the orthogonal constraint, compared to some other form of regularization - it would be nice for authors to elaborate on that.
>
> The biggest motivation behind orthogonal regularization lies in understanding Eq. (8) and (14) that the node features $\mathbf{H}$ becomes full rank matrix with good condition number.  Accordingly, the subspace that $\mathbf{H}$ can span can be rich enough. We plan to add an illustration that explains this geometric motivation behind orthogonal regularization.
>
>
> > 2. My biggest concern lies in the experimental portion of the paper. The authors do include results of ablation experiments in the supplementary materials, but in the proposed method is compared to only 1 other approach for the gender classification task and 2 variations of 1 approach for the decoding task. Furthermore, while it seems the result for the gender task was produced by the authors, the decoding results were taken from the literature, and it is unlikely that the exact same subjects and preprocessing step/experimental methods were used. Thus, I would say the numbers are informative but not directly comparable. Also, given that the ST-GCN result was produced by the authors, I don't know why this method wasn't also tested on the decoding task.
>
>
> We appreciate the important point raised by the reviewer.
> We are under implementing and experimenting with the baseline methods which could not have been fully completed within our response period due to resource constraints.
> We apologize for the inconvenience and will update the results as soon as they become available.
> For a preliminary comparison of the performance metrics before the final metrics are provided, we summarize the reported performance from the papers that used the HCP dataset.
>
> - Gender classification on HCP-Rest
>
> |      Method     |   FC    |   Accuracy   |   Reference |
> |---|---|---|---|
> |      *STAGIN     | dynamic | 88.20 (1.33) | Ours |
> |      ST-GCN     | dynamic | 83.7 ()      |  Gadgil (2020) |
> | TCN-GN-DiffPool | static  | 76 (1.0)     |  Azevedo (2020) |
> |       GIN       | static  | 84.61 (2.9)  |  Kim (2020) |
> |       GCN       | static  | 83.98 (3.2)  |  Arslan (2018) / Kim (2020) |
>
> We plan to implement other GNN models which have been applied to fMRI data but not for HCP-Rest gender classification.
>
> - Task decoding on HCP-Task
>
> |      Method     |         FC         |   Accuracy   | Reference |
> |---|---|---|---|
> |      *STAGIN     |       dynamic      | 99.19 (0.20) | Ours |
> |      BAnD++     | none (voxel-based) | 97.2 (0.57)  | Nguyen (2020) |
> |      BAnD       | none (voxel-based) | 97.0 (0.37)  | Nguyen (2020) |
> |      r-BAnD     |       dynamic      | 98.90 (0.27) | Ours for comparison |
> |      ST-GCN     |       dynamic      |              |  |
>
> We plan to implement ST-GCN on HCP-Task based on reviewers' suggestion.
>
>
> > 3.Authors appropriately list all the hyperparameter settings, and it seems that the same settings were used for all folds (no tuning). However, I am wondering about the setting for the time window length $\Gamma$. While authors give references to support their choice, in general window length in standard dynamic connectivity analysis is highly debated (actually as mentioned in one of their references) and the choice is generally arbitrary, with analysis results potentially greatly depending on the windowing. While the authors do provide a fairly complete study overall, I think it would be great to see the effect of different window sizes on the proposed method's results.
>
> Thank you for the important comment.
> We have performed experiments on different time windows and the results are as follows:
>
> |  $\Gamma$   |   Accuracy   |      AUROC      |
> |---|---|---|
> | 25 (18s) | 85.45 (3.51) | 0.9252 (0.0235) |
> | *50 (36s)| 88.20 (1.33) | 0.9296 (0.0187) |
> | 75 (54s) | 86.37 (1.87) | 0.9218 (0.0168) |
>
> Best performance was achieved with setting $\Gamma = 50$, which supports the conventional practice for selecting the window size of dynamic FC studies.

---

> ### Author Response · Authors · 2021-09-05
> **Response to Reviewer RkTU - Comparative experiments**
>
> We provide our comparative experiment results as follows.
>
> - Gender classification on HCP-Rest
>
> |      Method     |   FC    |   Accuracy   | AUROC  | # Parameters |
> |---|---|---|---|---|
> |      *STAGIN-SERO-40     | dynamic | 89.02 (1.80) | 0.9408 (0.0110)  | 1,209,804 |
> |      *STAGIN-SERO     | dynamic | 88.20 (1.33) | 0.9296 (0.0187)  | 1,209,804 |
> |      *STAGIN-GARO     | dynamic | 87.01 (3.00) | 0.9151 (0.0258)   | 1,068,426 |
> |      ST-GCN     | dynamic |  76.95 (3.00)    |   0.8545 (0.0316)     | 355,042 |
> | TCN-GN-DiffPool | static  | 53.70 (2.40)      |  0.5580 (0.0246)       | 849,507 |
> |       GIN       | static  | 81.34 (2.40)  |  0.8955 (0.0237)   | 169,996 |
> |       GCN       | static  | 80.79 (2.00)  |  0.8741 (0.0174)  | 101,896 |
> |       GraphSAGE       | static  |  75.48 (1.97)   |  0.8237 (0.0228)    | 202,248 |
> |       ChebGCN       | static  |  77.76 (2.09)  |  0.8582 (0.0233)    | 704,008 |
>
> Asterisks indicate our proposed models.
> STAGIN-SERO-40 refers to STAGIN-SERO with dynamic FC graph sparsity set to 40\%
>
> - Task decoding on HCP-Task
>
> |      Method     |         FC         |   Accuracy   | # Parameters |
> |---|---|---|---|
> |      *STAGIN-SERO     |       dynamic      | 99.19 (0.20) | 1,209,804 |
> |      *STAGIN-GARO     |       dynamic      | 99.02 (0.17) |  1,068,426 |
> |      BAnD++     | none (voxel-based) | 97.29 (0.46)  | 2,010,176 |
> |      BAnD       | none (voxel-based) | 95.77 (0.65) | 2,010,176 |
> |      r-BAnD     |       dynamic      | 98.90 (0.27) | 664,068 |
> |      ST-GCN     |       dynamic      | 98.92 (0.18) | 355,042 |
> |       GIN       | static  | 93.87 (0.66)   | 169,996 |
> |       GCN       | static  | 45.07 (1.63) | 101,896 |
> |       GraphSAGE       | static  |  54.52 (0.97)   | 202,248 |
> |       ChebGCN       | static  |  73.06 (0.68)   |  704,008 |
>
> It can be seen that static FC methods have difficulty in properly classifying the task types. The result is not very surprising since the sub-task timing information is completely lost in static FC methods, which can be a critical disadvantage in task classification.

---

> ### Author Response · Authors · 2021-09-13
> **Response to reviewer RkTU - Comparative experiments [2]**
>
> A new study was recently published as a preprint which attempted to apply MS-G3D to the resting-state fMRI data [1].
> MS-G3D is an extension of the ST-GCN model, developed for action recognition [2].
> We have re-experimented the model with the code provided by the authors (https://github.com/metrics-lab/ST-fMRI/), but revised the training phase to not early stop based on the test set which can significantly exaggerate the test performance.
> The model required 134,039,479 parameters to process ROI-timeseries generated from the Schaefer400 atlas which was over 110$\times$ the size of our proposed STAGIN-SERO, resulting in OOM with our same experimental settings.
> Accuracy of the MS-G3D on HCP-Rest resulted in $79.16 \pm 2.53 \%$ with ICA-extracted ROI-timeseries with 22 nodes.
>
> |      Method     |   FC    |   Accuracy   | AUROC  | # Parameters |
> |---|---|---|---|---|
> |      *STAGIN-SERO-40     | dynamic | 89.02 (1.80) | 0.9408 (0.0110)  | 1,209,804 |
> |      MS-G3D     | dynamic | OOM |  OOM | 134,039,479 |
> |      MS-G3D (22 nodes)     | dynamic | 79.16 (2.53) | 0.8912 (0.0329)   | 3,045,283 |
>
> We cautiously assume that the unexpectedly high accuracy reported in the paper can possibly be related to peeking of the test dataset during the training phase.
>
> [1] Dahan, Simon, et al. "Improving Phenotype Prediction using Long-Range Spatio-Temporal Dynamics of Functional Connectivity." arXiv preprint arXiv:2109.03115 (2021). \
> [2] Liu, Ziyu, et al. "Disentangling and unifying graph convolutions for skeleton-based action recognition." Proceedings of the IEEE/CVF conference on computer vision and pattern recognition. 2020.

---

### Decision · Program_Chairs · 2021-09-27

**Decision:**

Accept (Poster)

**Comment:**

The paper introduces a method for learning dynamic graph representation of brain connectome that
achieves state-of-the art performance while providing spatio-temporal explainability.
The work was appreciated by reviewers, as it put together novel tools (scaled dot attention / squeeze
excitation, self attention) with a graph neural network model to analyze dynamic graph data, specifically
4D fMRI. The application to the brain imaging data analysis was thought to be convincing. The reviewers
also consider that the paper is well-written.
However, it was found that
• not enough baselines had been used in applications, questioning the practical usefulness of the
method.
• some technical details were still somewhat arbitrary, questioning the generalizability of the method.
• comparisons against alternatives, in particular, attention-based methods, were not sufficient
Overall, the authors did a good work addressing the comments used, in particular regarding
comparisons/benchmarks against state of the art. However, identifying more difficult benchmarks would
indeed make the paper more convincing overall.
The final consensus is thus a weak accept.